# Absence of critical thickness for polar skyrmions with breaking the Kittel's law

Feng-Hui Gong ⬡[1,2,5], Yun-Long Tang ⬡[1,5], Yu-Jia Wang ⬡[1,5], Yu-Ting Chen[1,2,5], Bo Wu[3], Li-Xin Yang[1], Yin-Lian Zhu ⬡[1,3] ✉ & Xiu-Liang Ma ⬡[1,3,4] ✉

The period of polar domain ($d$) in ferroics was commonly believed to scale with corresponding film thicknesses ($h$), following the classical Kittel's law of $d \propto \sqrt{h}$. Here, we have not only observed that this relationship fails in the case of polar skyrmions, where the period shrinks nearly to a constant value, or even experiences a slight increase, but also discovered that skyrmions have further persisted in $[(PbTiO_3)_2/(SrTiO_3)_2]_{10}$ ultrathin superlattices. Both experimental and theoretical results indicate that the skyrmion periods ($d$) and $PbTiO_3$ layer thicknesses in superlattice ($h$) obey the hyperbolic function of $d = Ah + \frac{B}{h}$ other than previous believed, simple square root law. Phase-field analysis indicates that the relationship originates from the different energy competitions of the superlattices with $PbTiO_3$ layer thicknesses. This work exemplified the critical size problems faced by nanoscale ferroelectric device designing in the post-Moore era.

The Kittel's law ($d \propto \sqrt{h}$), derived from ferromagnetic materials, is also suitable for ferroelectric materials[1,2] (Supplementary Table 1), where $d$ is the period of polar domain, $h$ is the film thickness. Specifically, despite the fundamental differences of domain formation in ferromagnets and ferroelectrics, this square root expression corresponding to the domain periods and film thicknesses was then proved to persist in all these ferroics, according to the thermodynamic theoretical analysis[3,4].

In spite of its simplicity, this square root law holds over a remarkable range of sample sizes and shapes. It is then reasonable to think when or whether this law breaks down? Theoretically, two decades ago, Speck et al. have predicted a potential breakdown of the Kittel's law when approaching the ultrathin thicknesses of some ferroelectric and ferroelastic films. For example, in the studies of elastic energy and domain formation in the strained ferroelectric and ferroelastic films, the periods of domain will obey the Kittel's law in thicker films but will rises steeply when corresponding thicknesses shrink to a critical value[5–13], which is distinct from the description of the Kittel's law. Moreover, this breakdown was theoretically believed to

bring the ultrathin electronic devices with ultrahigh density storages possible[5–13], but never confirmed in experiments.

While these works are illuminative, polar topologies, such as the skyrmions and their distributions in ultrathin PbTiO₃ (PTO) layers, were not revealed. Particularly, powered by aberration corrected (scanning) transmission electron microscopy[14–25] ((S)TEM), the discovery and study of topological polar structures make them well suitable for further exploring the Kittel's law in the ultrathin limit[26–45]. Here, we directly observed this anomalous period-thickness relationship in polar skyrmion systems that could not be described by the simple Kittel's law, where the periods of skyrmions shrink nearly to a constant value, or even experiences a slight increasing in ferroelectric PTO/STO (SrTiO₃) superlattices with ultrathin PTO layer thicknesses. Moreover, low coercive voltage and good ferroelectricity were achieved in these ultrathin superlattices.

A series of $(PTO_n/STO_n)_{10}$ ($n = 37, 28, 23, 19, 12, 9, 4, 2$ u.c.) superlattices and $(PTO_n/STO_n)_1$ ($n = 50, 23, 12, 2$ u.c.) bilayers were deposited on STO substrates using pulsed laser deposition (PLD). Superlattice growth details are described in methods. It is of great

[1]Shenyang National Laboratory for Materials Science, Institute of Metal Research, Chinese Academy of Sciences, Wenhua Road 72, Shenyang 110016, China. [2]School of Materials Science and Engineering, University of Science and Technology of China, Wenhua Road 72, Shenyang 110016, China. [3]Bay Area Center for Electron Microscopy, Songshan Lake Materials Laboratory, Dongguan 523808 Guangdong, China. [4]Institute of Physics, Chinese Academy of Sciences, Beijing 100190, China. [5]These authors contributed equally: Feng-Hui Gong, Yun-Long Tang, Yu-Jia Wang, and Yu-Ting Chen. ✉e-mail: ylzhu@imr.ac.cn; xlma@iphy.ac.cn

importance to prove critical thickness of skyrmions for the designing of skyrmion based ultrathin and high-density information storage[46–56]. The polar skyrmion has a hedgehog-like component at the top and bottom interface between STO and PTO, where the polarization direction rotates smoothly from up to down and from the centre to the edge of the skyrmion. Moreover, a Bloch-like in-plane component forms at the central plane in PTO, where the polarization direction forms a closed loop (Supplementary Fig. 1a-c). The sizes of skyrmions may be modulated by PTO layer thicknesses (Supplementary Fig. 1b, c).

## Results and discussion

### Characterization of growth quality of PTO/STO superlattices

First we have employed STEM, X-ray diffraction (XRD) and atomic force microscopy (AFM) to verify the growth quality. Typical low-magnification STEM images and EDS mapping of the cross-sectional superlattices along the [010] zone axis reveal the sharp interfaces and good coherence (Supplementary Figs. 2, 3). The θ−2θ symmetric scan XRD patterns and surface topography also reveal that these super-lattices exhibit good epitaxy with atomic scale flat surface (Supplementary Fig. 4). These results suggest that we have obtained high-quality PTO/STO superlattices, which is important for the further revealing of the characters of possible polar skyrmions.

### Thickness-dependent skyrmions evolution

Using diffraction contrast analysis, selected electron diffraction (SAED) and XRD methods, we mainly focus on the critical thickness of polar skyrmions, that is, the critical PTO layer thickness could hold the formation of polar skyrmions and their statistical distribution features. For revealing these statistical details, a range of dark field (DF) images,

SAED patterns, reciprocal space maps (RSMs) corresponding to the serial $(PTO_n/STO_n)_{10}$ superlattices were performed (Fig. 1a–p and Supplementary Figs. 5, 6).

Figure 1a is a cross-sectional TEM DF image of the $(PTO_{37}/STO_{37})_{10}$ superlattices. Intensity modulations suggest the possible formation of polar skyrmions, which is similar to previous works[37]. The inset in Fig. 1a is a single enlarged (2̄02) SAED spot (full SAED pattern was shown in Supplementary Fig. 6a). The long-range ordering along the out-of-plane direction manifests the superlattice Bragg reflections, as the super-reflections along the out-of-plane direction shown in the enlargement SAED spot. Specially, satellite diffraction spots marked by two white arrows indicate some in-plane ordering modulations, which originate from polar skyrmions and is consistent with the DF image. To confirm the formation of polar skyrmions in the whole $(PTO_{37}/STO_{37})_{10}$ superlattices, (103) RSM map of the entire $(PTO_{37}/STO_{37})_{10}$ super-lattices was further performed (Fig. 1b). Here the satellite peaks were clearly recorded for the whole film sample, which suggest the obtained polar skyrmions undoubtedly. The in-plane modulation period could be calculated via the distance between the satellite peak and the Bragg peak

$$d = \frac{1}{\Delta q_x}, \tag{1}$$

which yields an in-plane period of 13.8 nm for the $(PTO_{37}/STO_{37})_{10}$ superlattices. In the same way, the DF images, single SAED spots and RSMs were shown in Fig. 1c–p for the rest of $(PTO_n/STO_n)_{10}$ (n = 28, 23, 19, 12, 9, 4, 2 u.c.) superlattices (corresponding full SAED patterns were shown in Supplementary Fig. 6b–g).

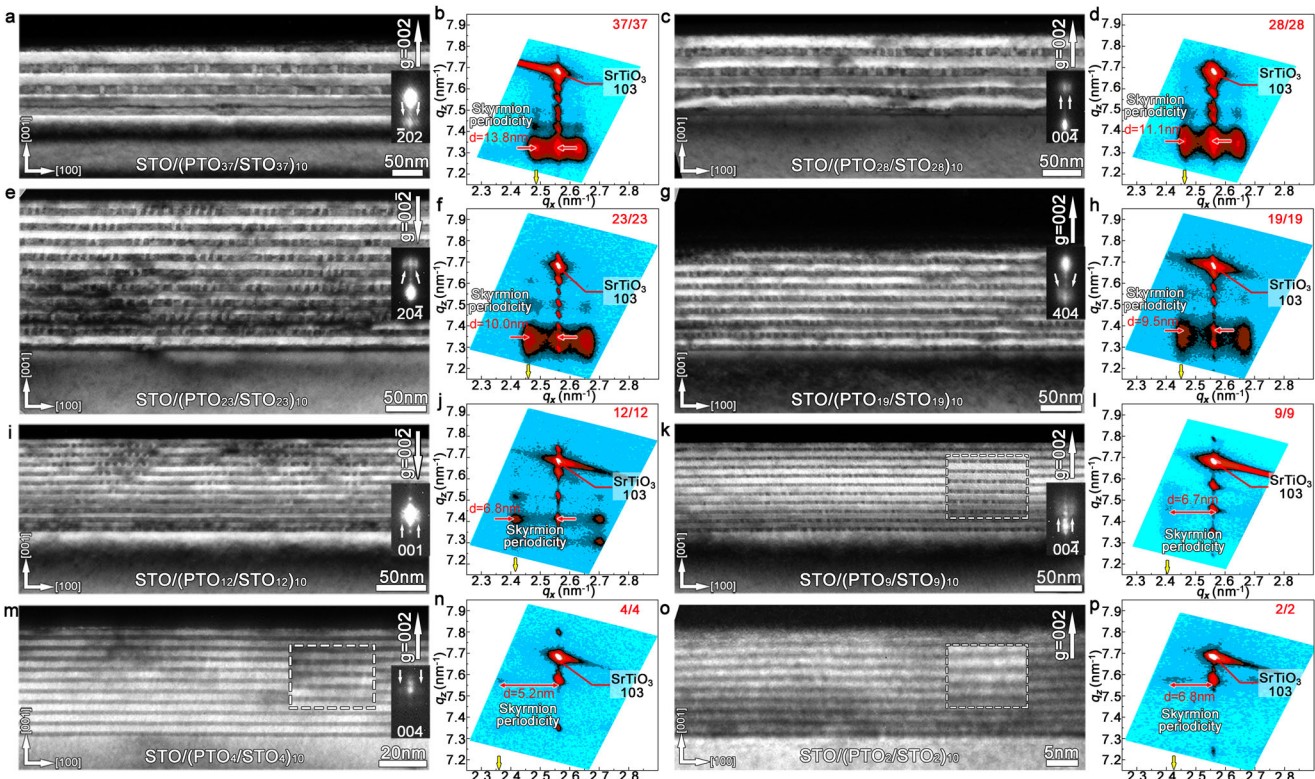

**Fig. 1 | Determination of the critical thickness for formation of polar sky-rmions. a–p** A series of cross-sectional DF images for $(PTO_n/STO_n)_{10}$ (n = 37, 28, 23, 19, 12, 9, 4, 2 u.c.) superlattices and their corresponding logarithmic RSMs in the $q_x$-$q_z$ scattering plane around the (103) substrates reflection. Note that the DF images taken by g = 002 or g = 002̄ diffraction vector. A range of enlarged single electron diffraction spots were inserted to the right of the DF images. The white dashed boxes in **k, m, o** are the partial enlarged view of the skyrmion images. For RSMs, the Bragg peaks (superlattice peaks) along the $q_z$ direction correspond to an out-of-plane periodicity of PTO/STO superlattices thickness, and satellite peaks along the $q_x$ direction correspond to an in-plane periodicity of polar skyrmions (13.8, 11.1, 10.0, 9.5, 6.8, 6.7, 5.2, 6.8 nm).

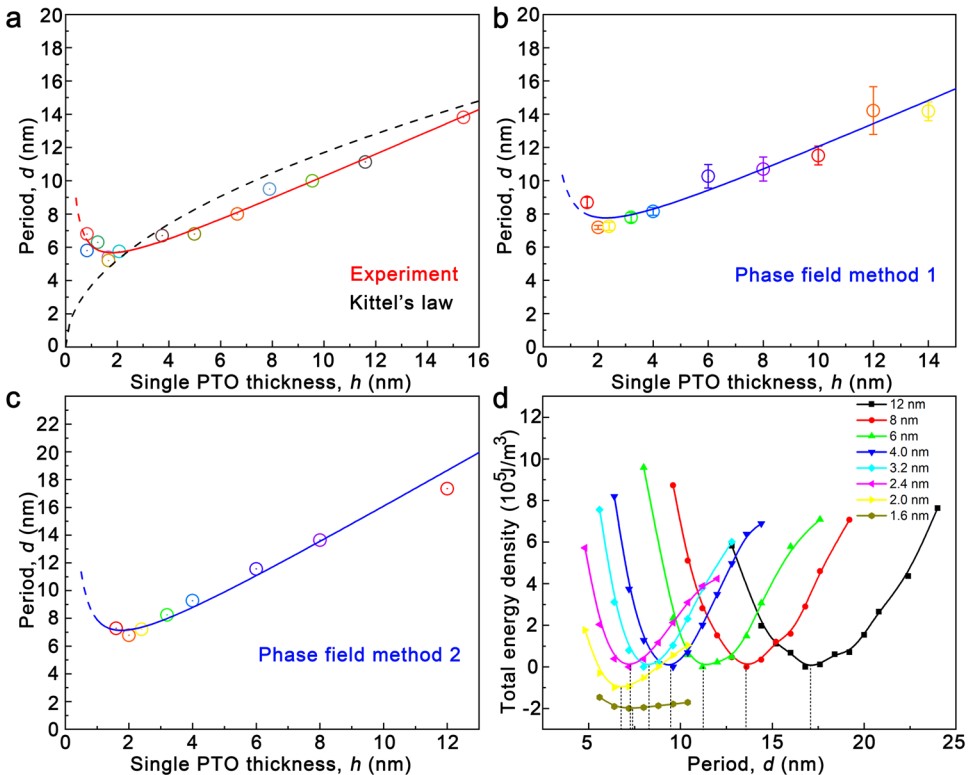

**Fig. 2 | Fitting the relationship between the period and thickness of skyrmions.**
**a-c** Fitted relationship between the periods ($d$) and the thicknesses ($h$) of polar skyrmions from experiments and simulations with two different methods. One experimental data point is from references[37]. The error bar in **b** comes from the statistics of several initial models, as can be detailed in the method section. **d** The total energy density curves for the STO/PTO/STO trilayers with different thicknesses in the skyrmion lattice models for the determination of the optimal periods in **c**. The lowest total energy density for each thickness was chosen as the reference. The solid curves and dashed drop lines are guides to the eye. For better visibility, the curves of 2.0 nm and 1.6 nm were shifted downward by $1 \times 10^5$ and $2 \times 10^5 \text{ J/m}^3$, respectively.

For the superlattices with thicker PTO layers ($n = 19$–$37$), they tend to obey the Kittel's law by that the in-plane modulation periods gradually decrease with the PTO layer thicknesses. However, for the superlattices with thinner PTO layers ($n = 2$–$12$), clear deviation of the Kittel's law could be glanced, as the in-plane modulation period for these samples keep almost a constant, which can be primarily identified as the unchanged $d$ in the corresponding RSMs (Fig. 1j, l, n, p and Supplementary Note 1). Here, importantly and unexpectedly, the satellite peaks also appear in the RSM of the $(PTO_2/STO_2)_{10}$ superlattices, which suggest the possible missing of critical thickness restriction for the formation of polar skyrmions in oxides, since here the PTO layer for maintaining polar skyrmions is even thinner than the critical thickness restriction for maintaining ferroelectricity in single layer PTO films[57–59].

**The period-thickness relationship of skyrmions**
By further detailing the statistical distribution of the in-plane modulation periods and PTO layer thicknesses, and combination of phase-field simulations, we revealed the relationship between the periods of polar skyrmions and the PTO layer thicknesses. The periods of polar skyrmions calculated via the SAED and RSM results were shown in Fig. 2a. The period of polar skyrmions are about 13.8, 11.1, 10.0, 9.5, 6.8, 6.7, 5.2 and 6.8 nm for the $(PTO_n/STO_n)_{10}$ ($n = 37$, 28, 23, 19, 12, 9, 4, 2 u.c.) superlattice series (Fig. 1a–p). Here it is clear that at the ultrathin PTO layer side, the period of polar skyrmions disobey the Kittel's law, where the periods shrink nearly to a nearly constant value, or even experience a slight increasing (The Kittel's law predicts a sharp drop of this period evolution, as the black dotted line indicate). Specifically, when approaching ~ 4 nm ultrathin PTO layer thickness, the periods of polar skyrmions will not further decrease with decreasing PTO thicknesses, where a constant value of about

5 ~ 6 nm could be maintained. In particular, even a slight increasing may be observed. These experimental data could be fitted by utilizing the Origin software (Fig. 2a, red solid line). The fitting function is

$$d = 0.68h + \frac{2.17}{h} + 3.24 \tag{2}$$

The first term describes the linear section (roughly $h > 2$ nm) and the second term dominates in the region of $h < 2$ nm, which is consistent with the feature of the experimental distribution.

We have performed two series of phase-field simulations to study the relationship between the skyrmion period and the PTO layer thickness and the results are presented in Fig. 2b, c. In Fig. 2b, the period was calculated by counting the number of skyrmions and in Fig. 2c, the period was determined by the local minimum of the total energy density, as shown in Fig. 2d. The complete calculation details can be found in the Method section. The two period-thickness curves in Fig. 2b, c show a decrease and then an increase of the period with the decrease of the PTO layer thickness, which is consistent with the experimental observation. The fitting results are

$$d = 0.72h + \frac{3.73}{h} + 4.48 \tag{3}$$

and

$$d = 1.32h + \frac{4.10}{h} + 2.48, \tag{4}$$

respectively.

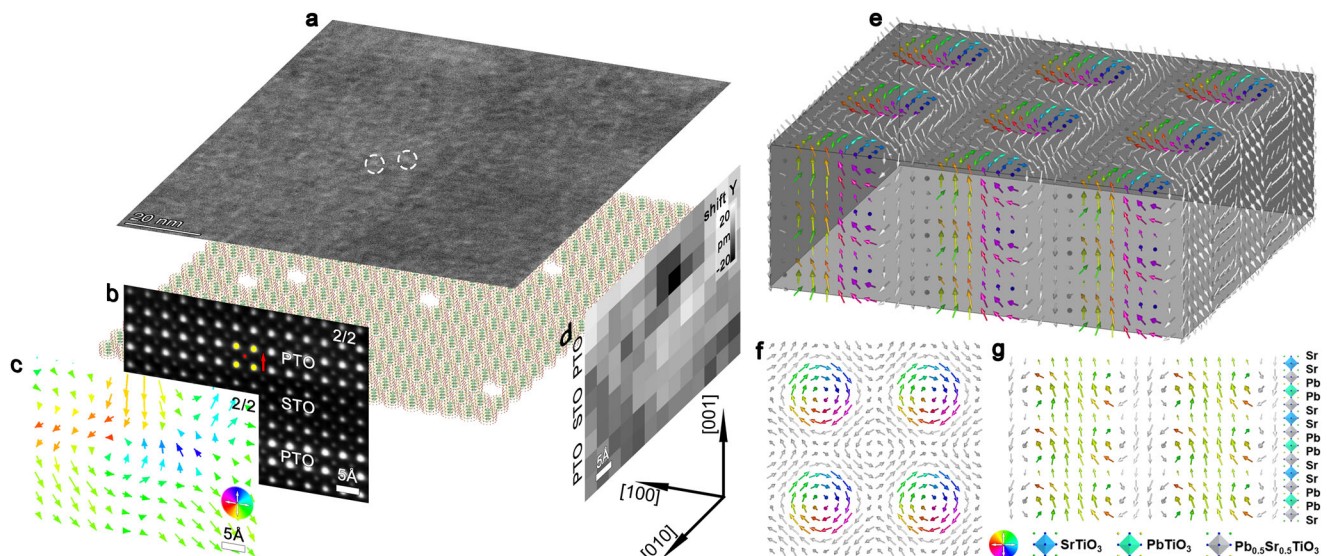

**Fig. 3 | Characterization of the limit polar skyrmion. a** Planar-view DF under-focus STEM imaging of $(PTO_4/STO_4)_{10}$ superlattices, showing the widespread occurrence of rounded polar skyrmions. The individual skyrmion is marked by white dotted circle. The 3D skyrmion maps are located under the under-focus STEM images. **b** Atomically resolved cross-sectional HAADF-STEM images of $(PTO_2/STO_2)_{10}$ superlattices. The yellow and red circles denote the Pb and Ti atom columns, respectively, and the red arrow denotes the direction of Ti-displacement ($\delta_{Ti}$) vectors. **c** The reversed $\delta_{Ti}$ vector maps (spontaneous polarization, Ps), revealing the thickness-dependent skyrmion. **d** The shift $Y$ along the out-of-plane direction. **e–g** The first-principles calculation result of a $PTO_2/STO_2$ superlattice containing a skyrmion. **e** The three-dimensional presentation of the polarization vectors. **f** The top view of the polarization vectors of the PTO layer showing the vortex-antivortex pattern. **g** The cross-sectional view of the polarization vectors of the front layer in **e**. The polarization vectors with positive $P_z$ components are colored by their in-plane directions and those with negative $P_z$ components are in the gray (or white) color. The spacing between vectors is one unit cell.

To further understand the period-thickness relationship, the energy components for several thicknesses were analyzed. Supplementary Fig. 7a gives the evolution of different energy components with the period for the 12 nm film. It is found that the energy components are all monotonous around the optimized period: The bulk and electrostatic energy densities increase with the period, while the gradient and elastic ones decrease. It is the competition of the sum of bulk and electrostatic energy densities and the sum of gradient and elastic energy densities that determines the optimal period. It should also be noted that the bulk energy density of the 12 nm film would experience a minimum with the decrease of period ($d = 13$ nm). As the decrease of the film thickness, the position of the bulk energy minimum gradually shifts to the optimal period, as shown in Supplementary Fig. 7b, c. The gradient energy density curve shows a maximum when the thickness is 3.2 nm, as shown in Supplementary Fig. 7c. The position of this maximum also shifts to the optimal period when the film thickness further decreases to 1.6 nm, as shown in Supplementary Fig. 7d. Comparing the four panels in Supplementary Fig. 7, it is found that when $h \geq 4$ nm, the energy components are all monotonous near the optimal period. When $h < 4$ nm, the bulk and gradient energy densities show a minimum and a maximum close to the optimal period, respectively. Since the monotonicity of different energy components are the prerequisite for the deduction of the Kittel's law, it is indicated that the bottom limit of the Kittel's law is $h = 4$ nm in this system. Thus, here we have summarized all thickness ranges for PTO layers where we believed that the period-thickness relationship in ultrathin films does not accord with Kittel's law.

It is well known that the depolarization field plays an important role in ultrathin ferroelectric layers[9,60]. Here, the electric field and polarization profiles along the film normal across the center of the skyrmion were plotted for several layer thicknesses, as shown in Supplementary Fig. 8. It is found that the polarization profiles are generally the same for different film thicknesses and the electric field profiles are obviously different for thicker and thinner films. When the film thickness is 20 nm, the electric field in the PTO layer, or the depolarization field, mainly concentrates near the PTO/STO interfaces and gradually decreases away from the interfaces. In the middle of the PTO layer, the depolarization field is nearly zero, as shown in Supplementary Fig. 8a. When the film thickness decreases, the depolarization field in the middle of the PTO layer gradually increases, as shown in Supplementary Fig. 8b–d. In other words, the effect of depolarization field becomes more important for thinner films. Other detailed discussion of the relationship can be found in the Supplementary Note 2.

## Observation of skyrmions in limit thickness superlattices

We further studied these polar skyrmions in thicker $(PTO_n/STO_n)_{10}$ ($n = 23$, 12 u.c.) superlattices as an example, via real space TEM imaging. Planar-view under-focus high-angle annular dark-field (HAADF) STEM images and atomic-scale polarization mappings using a displacement vector-mapping algorithm on cross-sectional HAADF-STEM images were performed to confirm the emergence of polar skyrmions in these PTO/STO superlattices (Supplementary Figs. 9–11, experimental details can be found in methods). The planar-view STEM images of the $(PTO_n/STO_n)_{10}$ ($n = 23$, 12 u.c.) superlattices (Supplementary Figs. 9a, 10a) display the widespread polar skyrmions and elongated polar skyrmions features along the [100] and [010] directions, which is similar to previously reported[37].

In addition, the polar skyrmions in the PTO/STO superlattices with thinner PTO layers was studied. The planar-view STEM image of the $(PTO_4/STO_4)_{10}$ superlattices also display the occurrence of polar skyrmions and elongated polar skyrmions features along the [100] and [010] directions (Fig. 3a, Supplementary Fig. 11f), which is similar to the PTO/STO superlattices with thicker PTO layers. The 3-dimensional (3D) schematics of the polar skyrmion configurations were shown in Fig. 3a and Supplementary Figs. 9a, 10a. Supplementary Figs. 9b, 10b are corresponding atomically resolved cross-sectional HAADF-STEM images. The $Pb^{2+}$ columns in these HAADF-STEM images appear as the brightest dots because the intensity of atom columns is approximately proportional to $Z^2$, where $Z$ is the atomic number[15]. The polarization direction is opposite to the direction of $\delta_{Ti}$ (Supplementary Figs. 12,

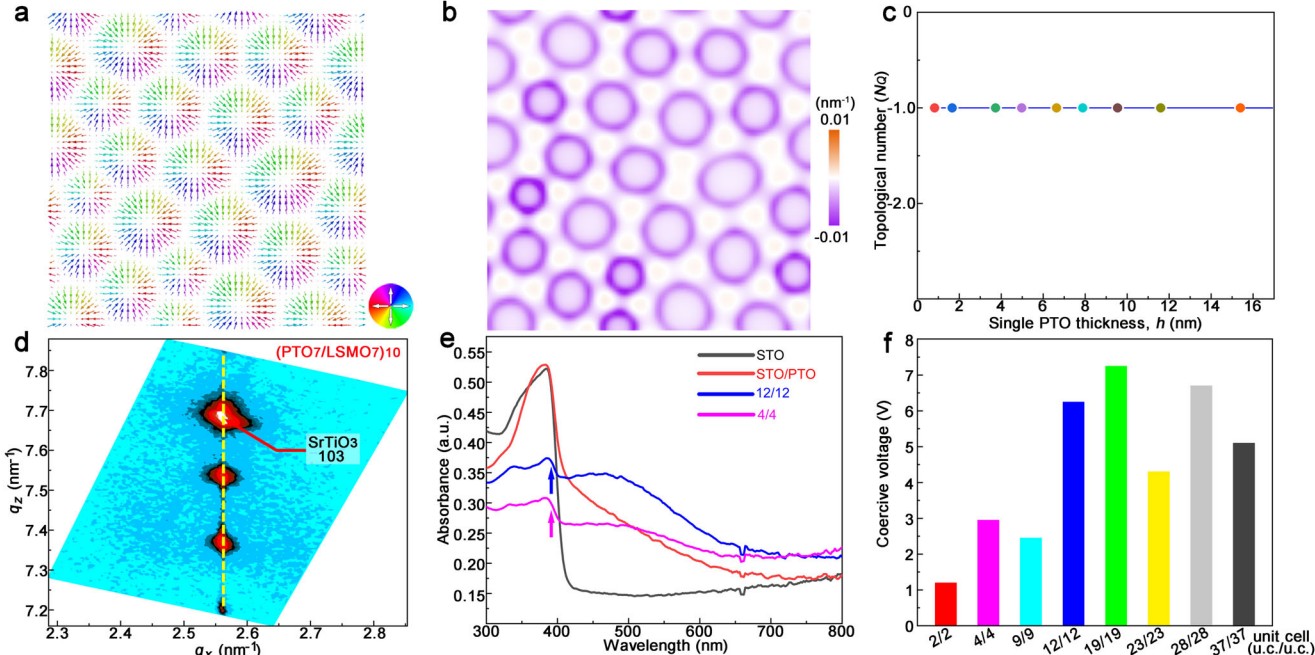

**Fig. 4 | The topological number and possible formation mechanism of ferroelectric skyrmions.** The distributions of polarization **a** and topological density **b** at the PTO/STO interface for the STO/PTO/STO model with the PTO layer thickness of 4 nm obtained from phase-field simulations. **c** Dependence of the topological number ($N_Q$) on thickness. **d** RSM around (103) Bragg spot of the STO substrate for the $(PTO_7/LSMO_7)_{10}$ superlattices. **e** UV absorption measurements on the $(PTO_n/STO_n)_{10}$ ($n = 12$, 4 u.c.) superlattices. A bare STO and single PTO film are also measured here for comparison. **f** Coercive voltage distribution map corresponding to different superlattice thicknesses.

13). Polarization maps (Supplementary Figs. 9c, 10c) based on the HAADF-STEM images (Supplementary Figs. 9b, 10b) indicate clear domain structures with anti-parallel polarization and polarization rotations in these PTO/STO superlattices, which is the feature of the polar skyrmions as also reported previously[37–39]. Furthermore, for the $(PTO_2/STO_2)_{10}$ superlattices, polarization map (Fig. 3c) based on the HAADF-STEM images (Fig. 3b) also indicates the signal of the polar skyrmions, which corresponds to the red dotted box in Supplementary Fig. 14. Figure 3d indicates that the ion displacement of closed ring along the out-of-plane direction is about 0 ~ 20 pm.

Supplementary Fig. 15a-d are the DF image of $(PTO_{50}/STO_{50})_1$ bilayer and planar-view HAADF-STEM images of $(PTO_n/STO_n)_1$ ($n = 23$, 12 u.c.) bilayers. Supplementary Fig. 16 are the cross-sectional and planar-view HAADF-STEM images of $(PTO_2/STO_2)_1$ bilayer. The polarization distributions of the white dotted areas, clearly show the skyrmion polar characteristic (Supplementary Figs. 15, 16). The planar-view polar map further proves that they are the skyrmions rather than other topologies (Supplementary Figs. 15, 16 and Supplementary Notes 3, 4).

Based on all the above results, the skyrmions could be confirmed in these PTO/STO superlattices including the ones with ultrathin PTO layers. Here the PTO layer for maintaining polar skyrmions (~ 2 u.c.) is even thinner than the critical thickness restriction for maintaining ferroelectricity (~ 3 u.c.) in single layer PTO films[57,58], thus we also deduced that the critical thickness of the polar skyrmion could be missing. Meanwhile, we also proposed a possible size shrinking mechanism based on 3D skyrmion evolution with decreasing superlattice thickness (Supplementary Fig. 14). To show that the skyrmions could exist in the ultrathin superlattices, an atomic model of $PTO_2/STO_2$ (The model size is $10 \times 10 \times 4$ u.c.) was constructed and relaxed by using first-principles calculations. Figure 3e-g give the 3D polarization structure and the plan-view and cross-section slices of the optimized model and the feature of skyrmion can be clearly revealed. Compared with the skyrmion in a larger model of $PTO_{10}/STO_4$ whose

size is $20 \times 20 \times 14$ u.c. predicted by second-principles calculations[37], the Bloch characteristic in the PTO layer is enhanced while the Néel characteristic at the PTO/STO interfaces is not very obvious.

The evolution process with thickness of polar skyrmions could be glanced in Supplementary Fig. 14. By comparing the skyrmion evolution processes with thickness, temperature and electric field[38,42], we found that they have similar evolutionary characteristics. In addition, we also performed in-situ TEM experiment to observe the dynamic evolution of polar skyrmions. Supplementary Fig. 17 shows a series of DF images of the $(PTO_{12}/STO_{12})_{10}$ superlattices from 423 K to 623 K. The phase transition temperature from the polar skyrmion to the paraelectric state, or the Curie temperature, could be regarded as about 623 K. When the temperature rises from 473 K to 573 K, the polar skyrmions disappear layer by layer as marked by the yellow dashed box, which implies that skyrmions have collective and cooperative behavior in the same PTO layer. We also studied the evolution of skyrmions in other thickness superlattices with temperature by experiments and phase-field simulations (Supplementary Figs. 18, 19). As shown in Supplementary Fig. 19, the averaged polarization magnitude decreases with the increase of temperature for all PTO layers considered and thinner films possess smaller Curie temperatures. The calculated Curie temperature of the 4 nm PTO layer is about 437 °C (710 K), close to the experimental result of the 4.8 nm PTO layer, 623 K.

## Calculation of topological numbers and functional properties

We proved that skyrmions can exist in ultrathin superlattices and explored the evolution behavior of skyrmions with thickness. The topological number was calculated based on the simulation results. Figure 4a, b give the polarization distribution at the PTO/STO interface and the corresponding topological density distribution. It is found that the polarization vectors form a convergent pattern and the topological density mainly concentrates at the periphery of each skyrmion. The skyrmions are characterized mathematically by the topological

number,

$$N_Q = \frac{1}{4\pi} \int\int \mathbf{u} \cdot \left(\frac{\partial \mathbf{u}}{\partial x} \times \frac{\partial \mathbf{u}}{\partial y}\right) dxdy, \qquad (5)$$

where **u** represents the normalized dipole moment, the surface integral is taken over the (001) plane. By integrating the topological density, it is found that the topological charges of these skyrmions are all −1. The topological charge of a skyrmion is related with the polarization direction of the skyrmion center: If the polar vector points to the +z direction, the topological charge is +1, and if it points to the −z direction, the topological charge is −1. As the thickness decreases, the topological number corresponding to the surface integral is always −1, as confirmed by the simulation results (Fig. 4c). Moreover, we further explored the effect of electric boundary conditions on polar skyrmions and its relevant functional properties. The skyrmion satellite peak disappears after the STO layer is replaced by LSMO ($La_{0.7}Sr_{0.3}MnO_3$) electrode, which indicates that the formation of skyrmion is significantly affected by the electric boundary condition (Fig. 4d, Supplementary Figs. 20, 21 and Supplementary Table 2). The formation of polar vortices maybe originates from Dzyaloshinskii−Moriya-like (DMI) interaction[26,61], which is not significantly affected by electric boundary conditions, such as $La_{0.7}Sr_{0.3}MnO_3$ and $SrRuO_3$ (Supplementary Table 2). By comparison, polar skyrmions are significantly affected by electric boundary conditions. The absence of polar skyrmions in the PTO/LSMO superlattices may be associated with charge screening at the interface which may alter the depolarization fields signifying the importance of electrostatic boundary conditions. On the contrary, the ferroelectric layers sandwiched by insulators may result in a large depolarization field at the film interface, which will become even dominant in ultrathin layers and affect classic Kittel's law[62].

Next, we have performed UV absorption measurements on the $(PTO_n/STO_n)_{10}$ (n = 12, 4 u.c.) superlattices grown on STO substrate to verify how the skyrmions changes optical property (Fig. 4e). We can find that the single crystal STO and single PTO film display steep absorption edge (around 400 nm). Compared with pure STO and PTO, the drop interval of the absorption edge for the superlattices decreases, as indicated by the arrow. The superlattices containing skyrmions show more intensive absorption over the whole visible range of wavelengths investigated. This may be derived from the complex hedgehog-like and Block-like polarization component in skyrmions PTO.

The ferroelectric responses of these ferroelectric/paraelectric superlattices were further studied by piezoresponse force microscopy (PFM) methods (Fig. 4f and Supplementary Fig. 22). The local switching behaviors of the $(PTO_n/STO_n)_{10}$ superlattices under applied voltage are given in Supplementary Fig. 22 (n = 37, 28, 23, 19, 12, 9, 4, 2 u.c.). The local piezoresponse amplitude loops along out-of-plane direction exhibit the classic butterfly shape. The local square phase hysteresis signals indicate clear switching behavior which further illustrates corresponding ferroelectricity nature. The asymmetric characteristic of the PFM loops may be attributed to the different work functions between the PFM conducting tip and STO substrate. Furthermore, we found that the coercive voltage of the superlattices reflected here tend to gradually decrease as the PTO layer thicknesses decrease (Fig. 4f, the coercive voltage calculation can be found in methods). The coercive voltage is about 1.2 V for $(PTO_2/STO_2)_{10}$ superlattices. In addition, Supplementary Fig. 22i is the out-of-plane PFM phase map (IMR area) of the $(PTO_{19}/STO_{19})_{10}$ superlattices after applying + 15 V. Here the IMR area was initially scanned at a bias of + 15 V, then remaining area was scanned at a bias of − 15 V. Supplementary Fig. 22j, k are the out-of-plane PFM phase and amplitude (3 × 3 μm²) maps of the $(PTO_{12}/STO_{12})_{10}$ superlattices, which was scanned after applying + 10 V. The 180° phase contrast reversal in these PFM maps also depicts the switchable characteristics. Above all, these experiments indicate that the superlattices exhibit good ferroelectricity and polarization reversal properties.

This work has demonstrated that the critical thickness for the formation of polar skyrmions may be missing. Polar skyrmions can be maintained in ultrathin superlattices and bilayers with PTO layer as thin as two unit cells, which will be important for the future study of skyrmion-based ferroelectric physics and related electronic elements. The skyrmion periods (d) and superlattice thicknesses (h) disobeys the Kittel's law, especially for the samples with ultrathin PTO layers. Our results indicate that the critical thickness for polar topologies may be even thinner than the maintaining of ferroelectricity in ultrathin ferroelectric films up to even nearly vanishing, which will be important for designing novel electronic devices in the post-Moore era, such as ultrahigh density memories.

## Methods

### Sample preparation details

The $(PTO_n/STO_n)_{10}$ (n = 37, 28, 23, 19, 12, 9, 4, 2 u.c.) and $(PTO_7/LSMO_7)_{10}$ superlattices were deposited on STO (100) substrates by PLD, using the Coherent ComPex PRO 201 F KrF excimer laser (λ = 248 nm). The $(PTO_n/STO_n)_1$ (n = 50, 23, 12, 2 u.c.) bilayers were also deposited on STO (100) substrates. In addition, the $(PTO_4/STO_4)_{10}$ superlattices was also deposited on Fe0.05% - doping STO (100) substrates. The sintered ceramic PTO target with 3 mol% Pb-enriched was used for PTO layer deposition. Single crystal STO target was used for STO layer deposition. The sintered ceramic LSMO target was used for LSMO layer deposition. Before deposition, the STO substrate was pre-heated to 750 °C to clean the surface of substrate and then cooled down to 700 °C for super-lattices growth. The PTO, STO and LSMO targets were pre-sputtered for 2000 laser shots. During deposition, the target-substrate distance was 32 mm, the oxygen pressure was 10 Pa, laser energy was 370 mJ, and repetition rate was 4 Hz. After deposition, these samples were annealed at 700 °C for 5 min and then cooled down to 200 °C with a cooling rate at 5 °C·min⁻¹ in an oxygen pressure of $3 \times 10^4$ Pa.

### High-resolution RSM

A Panalytical X'Pert Pro X-ray Diffractometer with a copper source was used for superlattices. The X-ray wavelength from the copper source was 1.54059 Å. RSMs were performed using a high-resolution XRD (A Bruker D8 advance high-resolution X-ray Diffractometer). Symmetric XRD scan with appropriate angle range was adopted to reveal the high crystalline of the superlattices and the nature of the interfaces. The (103) diffraction spot contains both in-plane and out-of-plane information. The polar skyrmions are the reproducible for (103) RSM of the $(PTO_n/STO_n)_{10}$ (n = 4, 2 u.c.) superlattices grown on STO substrate.

### TEM and STEM

Cross-sectional samples for TEM and STEM observation were prepared by slicing, gluing, mechanical grinding, dimpling, and finally ion milling. The samples were dimpled down to <15 μm. Then a Gatan Precision Ion Polishing System (PIPS) 691 or 695 was used for ion milling. At the beginning of ion milling, the incident angles of 8° and milling voltage of 6 kV were used. Then, the angles and voltage were gradually reduced to 4° and 3 kV. Finally, ion milling voltage was set at 0.8 kV (691) or 0.1 kV (695) to reduce amorphous layer produced by ion beam damage. Unlike the cross-sectional samples, the plane-view sample was milled only from the substrate side to protect the film from damage. Here, as also indicated by our phase-field calculations, the effect of electric boundary conditions on the formation of skyrmions is much greater than that of strains. Thus the TEM sample preparation will not change the formation of the skyrmions in the superlattices grown on STO. The specimens were cleaned before they were allowed to be inserted into the TEM. The DF images were acquired by Tecnai $G^2$ F30 TEM. SAED patterns were recorded using JEOL 2010 high-resolution TEM. HAADF-STEM

images were acquired using a Titan G$^2$ 300 kV microscope with a high-brightness field-emission gun and double aberration correctors from CEOS. The beam convergence angle is 25 mrad, and collection angle is 50-250 mrad. Each of the HAADF-STEM images was acquired by adding up 20 original images, recorded from the same area of the superlattices. Image acquisition and processing were performed using the Velox software (FEI). Acquiring images in this way can reduce the influence of sample drift and scanning noises.

## Peak finding

For accurately distinguishing between A-site atoms and B-site atoms according to the intensity of the atomic column in the HAADF-STEM image, the brightness and contrast of images have been adjusted as a whole. The positions of atom columns in HAADF-STEM images were determined accurately on the basis of the 2D Gaussian fitting, which was carried out using the Matlab software[63]. The lattice spacing and Ti$^{4+}$ shift ($\delta_{Ti}$) were deduced.

## Phase-field simulation

The polarization distribution in the PTO/STO superlattices was studied by phase-field simulations. The evolution of polarizations is governed by the time-dependent Ginzburg-Landau equation:

$$\frac{dP_i}{dt} = -L\frac{\delta F}{\delta P_i} \qquad (6)$$

where $L$ is the dynamic coefficient and $F$ is the system's free energy which is composed of bulk, gradient, elastic and electrostatic ones:

$$F = \int_V [f_{bulk}(P_i) + f_{grad}(P_{i,j}) + f_{elas}(P_i, \varepsilon_{kl}) + f_{elec}(P_i, E_i)]dV \qquad (7)$$

The base area of the simulation box is $40 \times 40$ nm$^2$. In the thickness direction, one PTO layer sandwiched by two STO layers were stacked, to mimic the PTO/STO superlattices. The PTO and STO layers are in the same thicknesses. The thickness of a single PTO or STO layer varies from 1.6 nm to 14 nm. Since the zero strain states of PTO and STO correspond to the lattice parameters of 3.957 Å and 3.905 Å, respectively, different misfit strains (−1.3% and 0.0%) were applied to the PTO and STO layers, respectively. A 4 nm substrate layer was also considered in the solution of the mechanical equilibrium equation. The top surface is in a traction-free state, while the bottom surface of the substrate layer is fixed to the substrate strain. The short-circuit electric boundary condition is applied to the top and bottom surfaces of the model. Other technical details and the material parameters of PTO and STO can be found in previous literatures[17,64,65]. Two methods were adopted to study the period-thickness relationship. In the first one, the in-plane size of the simulation box is fixed to be 40 nm and several initial models with different random noises were used to produce the randomly distributed skyrmions. The approximate period for each model was calculated by

$$d = \sqrt{S/N} \qquad (8)$$

where $S$ is the base area of the simulation box ($40 \times 40$ nm$^2$) and $N$ is the number of skyrmions. The final period is the average period of these models. In the second method, the skyrmion lattice models with varying periods (or the in-plane sizes) were considered and the optimized periods were determined by the local minima of the total energy densities. For each STO/PTO/STO trilayer, there exists a minimum in the energy density-period curve. We have used a series of polynomials to fit these curves to obtain the optimal periods and found the cubic polynomial can fit all these curves well.

## First-principles calculations

The atomic relaxations were performed by Vienna ab-initio simulation package (VASP)[66]. The PBE-sol exchange-correlation functional[67] was used with the method of projector augmented-wave[68]. The energy cutoff was chosen as 500 eV. O's 2s2p, Ti's 3s3p3d4s, Sr's 4s4p5s and Pb's 5d6s6p electrons are treated as the valence electrons. Only the Gamma point was considered in the K-space. The ionic relaxation was considered as convergent when the force on each ion is <10 meV/Å. The size of this model is $10 \times 10 \times 4$ unit cells (2000 atoms). The initial model contains a square reversed domain with the lateral length of 6 u.c.. At the domain walls, the Bloch-type polarization components were assigned in a vortex pattern. After the atomic relaxation, the size of the reversed domain shrank to a round shape, as shown in Fig. 3f. The polarization vector of each unit cell was calculated according to the Born effective charge tensors of all atoms[69], which were obtained by a static calculation of a $1 \times 1 \times 4$ supercell of PTO$_2$/STO$_2$ where all the atoms were in their cubic positions.

## In-situ study

In-situ experiment was carried out using a Tecnai G$^2$ F30 TEM. The real-time DF TEM images were recorded using a double-tilt holder with heating function.

## Piezoelectric force microscopy (PFM)

Before the PFM test, the conductive silver glue was applied to the iron sheet, the substrate side of the sample was glued to the iron sheet, and the sample was heated to 60 °C and baked for 20 min. Surface morphology, the domain writing, hysteresis loops and out-of-plane amplitude/phase images were collected using the lithography PFM and the Dual AC Resonance Tracking (DART) mode at room-temperature (Cypher, Asylum Research). A DART model can relieve the topographic characteristics interference. Conductive Ti/Ir-coated silicon cantilevers (Asylum Research, ASYELEC-01-R2) were used for PFM measurements[70]. The typical tip radius is about 7 nm and the force constant is ∼ 2 N m$^{-1}$. The coercive voltage is calculated from the equation:

$$U = \frac{|U_+| + |U_-|}{2} \qquad (9)$$

where $U_+$ and $U_-$ represent the voltage corresponding to the lowest point in the amplitude loops, respectively.

## Reporting summary

Further information on research design is available in the Nature Portfolio Reporting Summary linked to this article.

## Data availability

The data that support the findings of this study are available within the article and the Supplementary Information. The presented data were available from the corresponding author upon request.

## Code availability

The computer code that supports the findings of this study is available from the corresponding authors upon reasonable request.

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

## Acknowledgements

We are grateful to Chang-Ji Li at Shenyang National Laboratory for Materials Science for experimental assistance with XRD and RSM. Hua-Long Ge and Ming Lv provided support for in-situ TEM. B.W. and L.-X.Y. contributed technical support on the Titan platform of the $G^2$ 60-300 kV aberration-corrected STEM. We are grateful to Jie-Feng Cao, Fang-Yuan Zhu, Ya-Mei Wang and Rui Yu at BL07U beamline and Xing-Min Zhang at BL02U2 beamline of Shanghai Synchrotron Radiation Facility for experimental assistance. This work is supported by the National Natural Science Foundation of China (51971223 (Y.-L.Z.), 51922100 (Y.-L.T.), 52122101 (Y.-J.W.)), the Key Research Program of Frontier Sciences CAS (QYZDJ-SSW-JSC010 (X.-L.M.)) and Shenyang National Laboratory for Materials Science (L2019R06 (X.-L.M.), L2019R08 (Y.L.Z.), L2019F01 (Y.-L.T.), L2019F13 (Y.-J.W.)). Y.-L.T. acknowledges the Scientific Instrument Developing Project of CAS (YJKYYQ20200066). Y.-L.T., Y.-J.W. acknowledge the Youth Innovation Promotion Association CAS (Y202048, 2021187).

## Author contributions

Y.-L.Z. and X.-L.M. designed the experiments. F.-H.G. and Y.-L.T. conceived the project of interfacial characterization in oxides by using aberration-corrected STEM. Y.-J.W. conducted the phase-field modeling, the first-principles calculations and the digital analysis of the STEM images. F.-H.G. and Y.-T.C. performed the thin-film growth. B.W. and L.-X.Y. participated in the thin-film growth and STEM imaging. F.-H.G. and Y.-T.C. carried out the scanning probe-based PFM measurements. All authors contributed to the discussions and manuscript preparation.

## Competing interests

The authors declare no competing interests.
