## [Peer Review File · Nature Communications]

Absence of critical thickness for polar skyrmions with breaking the Kittel's lawREVIEWER COMMENTS

Reviewer #1 (Remarks to the Author):

The authors report scaling laws of skyrmion lateral size versus film thickness. They found that there are three regimes where show different power laws. Especially, the size linearly depends on thickness in the intermediate thickness regime while the size can be reciprocally proportional to thickness when thickness is less than ~ 2 nm. The extensive TEM works on a series of the superlattices were made. The analysis results with simulation are also highly useful in understanding the topological structures. The following questions should be answered to be further considered with improving the manuscript.

1. Although the phase field simulation can explain the scaling behaviors and the sub-components of the free energy are analyzed. It is hard to catch up why the linear and inversely linear behaviors come out in the two-different thickness regimes. As the conventional Kittel's law can be understood as a consequence of the competition between bulk energy and wall energy, more simplified model equations might be proposed based on the quantitative information obtained from the phase field model.
2. The planar view DF images (e.g., Fig. 3a) are difficult to identify the detailed structure. Authors argue the topological number is 1 as presented in Fig. 4b. It is not clear which area they calculate over. As an example, it is necessary to plot an enlarged view of planar dipole distribution and the corresponding skyrmion density map.
3. How can we understand the ultrathin thicknesses result in topologically non-trivial textures in (001) plane different from the thicker ones?
4. Many of the macroscopic characteristics in Fig. 4 seem to be not directly related to the main theme of the paper.

Reviewer #2 (Remarks to the Author):

The manuscript reports an interesting phenomenon that the skyrmion periods and superlattice thicknesses disobey Kittel's law in PTO/STO superlattices with ultrathin PTO layers. By using the advanced PLD and TEM techniques, the authors show extremely high-quality results, including STEM, EDS, polar map, and RSM images. The data present in the manuscript are excellent and convincing. The theoretical simulation also gives a reasonable explanation of the experimental phenomenon.

The breakdown of Kittel's law was predicted 20 years ago. I believe this discovery in ferroelectric topological domains is essential to research aiming at ferroelectric ultra-high density storage. In addition, due to its extremely small size and interesting physical properties, the polar skyrmion has received extensive attention since it was first discovered in 2019. I am sure this finding has great potential merits to be published in Nature Communication. Before publication, the paper still needs a minor modification and raises the following questions and concerns as listed.

1. The polar skyrmion is a particle-like polar topology. Can "in-plane modulation periods" in RSM of cross-sectional samples fully represent the d (the periods of polar domain) in Kittel's law? Some direct evidence from the planar view might be considered, which helps to increase credibility. For example, adding some planar view STEM images and corresponding density statistics of polar skyrmions in typical systems, PTO_n/STO_n superlattices. We do directly see the density distribution of the skyrmions get increase as the PTO gets thinner, in Supplementary Fig 2. But, is it directly related to the periods of polar

domain?

2. I am wondering if the topological domains in superlattice with thicker PTO layers are still kept as polar skyrmions? Maybe due to the image grayscale, it is hard to see obvious circular or elongated features in Extended Data Fig. 7 and Supplementary Fig. 2a. I suggest the author should add a mark on the polar domains for evaluating the size. The data of the cross-sectional view is very clear and detailed; If adding some data of the planar view (polar map or PFM), which will further prove that they are the larger skyrmions rather than other topologies.

3. In Fig 3c, does the polar topology constructed in both STO and PTO layers? In addition, the cross-section view has the feature of a vortex structure, is there any possible a new polar type of topology?

4. I am personally interested in the calculation of topological numbers in Figure 4b. Are these calculated by simulation results in Supplementary Fig. 5, or by the polar maps obtained from experiments? What exactly does the “as confirmed by the experiment” mean?

In conclusion, I recommend this work for publishing in Nature Communication after a minor modification.

Reviewer #3 (Remarks to the Author):

“Absence of critical thickness for polar skyrmions with breaking the Kittel’s law” by F. Gong et. al

Ultrathin ferroelectric devices are important for high-density data storage applications and other nano-electronic technologies. However, the depolarization field and interface effects in thin films may become dominant at reduced thicknesses which significantly affect the domain structure and switching properties of ferroelectric materials. The availability of advanced thin film growth and structural imaging technology has boosted the research on dimensionally confined ferroelectrics which can play an essential role in developing a fundamental understanding of non-conventional behavior observed at reduced dimensions in ferroelectrics. In this work, Gong et al. have studied the effect of layer thickness on the domain structure in PbTiO₃/SrTiO₃ superlattices down to two unit cells (u.c.). By using X-ray diffraction and scanning transmission electron microscopy, the authors confirmed the formation of polar skyrmions in the heterostructures which could be retained at an ultralow thickness of the PbTiO₃ layer (2 u.c.). The authors particularly observed the deviation from Kittel’s law (which relates the domain width or domain period with the square root of film thickness) at these reduced dimensions. This deviation or ‘breaking’ from Kittel’s law is supported and explained based on phase-field calculations. Polarization switching and the effect of electric boundary conditions have also been studied in ultrathin layers. While the techniques used in this work are state-of-the-art and reliable, the subject matter lacks sufficient novelty, and the discussion lacks adequate depth and insight to warrant publication. The following are some aspects that need further improvement before the manuscript could be considered for publication:

1) The novelty of this work is unclear. The breaking from Kittel’s law at ultralow thicknesses has been theoretically predicted and experimentally shown in previous works (J. Appl. Phys., 1993, 74(10), 6012-6019 also ref. 5, Ferroelectrics, 1997, 202(1), 267-274 also ref. 8 & J. Appl. Phys, 2008, 104(6), 064109).

2) The type of medium immediate to the ferroelectric layer (vacuum/metal/insulator) plays an important role in determining the electrostatic boundary conditions. In the current work, the ferroelectric layers are sandwiched between insulators which may result in a large depolarization field at the film interface that becomes dominant in ultrathin layers and affect

the breakdown of Kittel's law. Previously developed models for the sub-Kittel regimes provide insight into the electrostatic description of ultrathin ferroelectric layers in different settings (Ferroelectrics, 1997, 202(1), 267-274 & R. Soc. Open Sci., 2020, 7(11), 201270). In the current work, the relationship between film thickness and the period of polar skyrmions has only been empirically established while the discussion based on phase-field simulation results seems rudimentary. It is suggested to provide further discussion taking into account the electrostatic boundary conditions, especially the depolarization field, and provide insight into the correlation between layer thickness and domain modulation. The fitting expression and parameters used in Fig. 2a need to be further analyzed keeping in view the previous models for sub-Kittel regions.

3) How do the authors define critical thickness here? Is it the layer thickness below which the domain period breaks from Kittel's law or below which the ferroelectricity disappears?

4) Based on HAADF and EDS images in Extended Fig 1 and 2 respectively, both the PbTiO₃ and SrTiO₃ layers in (PTO₂/STO₂)₁₀ superlattice seem to be 3 u.c. thick rather than 2 u.c. Can the authors explain how this superlattice is determined to be (PTO₂/STO₂)₁₀?

5) Compared to (PTO₁₂/STO₁₂)₁₀ superlattice, the evolution of domain structure in (PTO₂/STO₂)₁₀ superlattice at high temperatures would be more interesting since this may provide some insight into the order parameters driving the stability of skyrmions in ultrathin layers.

6) The effect of using LSMO as well as UV absorption behavior has been studied, however, it is not obvious how the obtained results support the main discussion or conclusion of the manuscript. For example, the absence of polar skyrmions in the PTO/LSMO superlattice may be associated with charge screening at the interface which may alter the depolarization fields signifying the importance of electrostatic boundary conditions.

7) It is not very obvious how the out-of-plane PFM measurements were obtained without a conductive bottom electrode or proper grounding since the substrate is insulative. Authors may add more information regarding sample configuration for the PFM measurement.

8) It is stated in line 325 "Our results indicate that the critical thickness for polar topologies may be even thinner than the maintaining of ferroelectricity in ultrathin ferroelectric films up to even nearly vanishing, which will be important for designing novel electronic devices in the post-Moore era, such as ultrahigh density memories". Although the individual layer of PbTiO₃ in (PTO₂/STO₂)₁₀ superlattice is 2 u.c. thin, the overall thickness of the film is still >15 nm which is much thicker than other ferroelectric thin films reported in the literature with switchable polarization. In this regard, will the polar skyrmions and ferroelectricity sustain in a single 2 u.c. thin PbTiO₃ layer sandwiched between two SrTiO₃ layers? Based on this, can authors further elaborate what is the general applicability of the results obtained in this work and how these superlattices may help in designing ultrahigh-density memories or cite relevant literature?

Reviewer #4 (Remarks to the Author):

This article explores the nature of the ferroelectric domain sizes in PbTiO₃/SrTiO₃ superlattices. Through probes of the domain size as a function of layer thickness, this paper concludes that there is no critical thickness and that the scaling does not follow the simple expectations of Kittel's law.

In this form, this paper is not suitable for publication in Nature Communications. There is a

major flaw in the conclusions that makes the questioning of thickness dependence invalid.

The authors make the statement that the samples show skyrmion order for all thicknesses, but the data shows that is not true. The skyrmion order has a distinct reciprocal space pattern as shown in previous work. While this is a (103) peak and not (003) as shown in references 30,31. The large-scale twisting nature of the skyrmion leads to broad peaks in reciprocal space which is maybe the case for the 9/9 superlattice but the signal to noise is very poor. The other thicknesses show very different satellite patterns likely from planar Ferroelectric phases (e.g. A1/A2), A/C order (12/12), and perhaps vortex phases. Regardless the satellite structures are evolving with thickness meaning that the real-space configuration is drastically changing. Kittel's scaling applies only in the case where the order is constant and is only changing in size. Likely these are mixed phase samples with complex heterogeneous order. An additional note is that the first order satellite is often not sufficient for complex characterization since higher order ones contain more detail concerning the structure. A side note on TEM, it is also not clear that there is skyrmion order present everywhere. TEM only samples a tiny fraction of the sample and is known to change the strain conditions.

As shown in previous work and the phase-field work(e.g. Small Methods 6, 2200486 (2022)), the phases are strongly dependent on the superlattice period and the strain state. This system is very unlikely to show the consistent order that can be used for a scaling study. Without a complex understanding of what phases are present and how they are mixed, one cannot try to apply a simple scaling argument.

Reply to reviewers' questions and comments:

Ref number: NCOMMS-22-42842-T

Title: Absence of critical thickness for polar skyrmions with breaking the Kittel's law

Authors: Feng-Hui Gong, Yun-Long Tang, Yu-Jia Wang, Yu-Ting Chen, Bo Wu, Li-Xin Yang, Yin-Lian Zhu & Xiu-Liang Ma

20 February, 2023

Reply to reviewer 1 (R1):

We appreciate the positive summary by the reviewer that “the authors report scaling laws of skyrmion lateral size versus film thickness”, “they found that there are three regimes where show different power laws”, and that “especially, the size linearly depends on thickness in the intermediate thickness regime while the size can be reciprocally proportional to thickness when thickness is less than ~ 2 nm.”. We also appreciate the positive comments raised by the reviewer that “the extensive TEM works on a series of the superlattices were made”, and that “the analysis results with simulation are also highly useful in understanding the topological structures.”. In the meanwhile, the reviewer also raises some specific questions. We understand the reviewer's concerns, and here we try to address all the questions and discuss all the comments one-by-one in the following. The revisions were indicated in **RED** characters in the revised manuscript.

Comment (R1.1): Although the phase field simulation can explain the scaling behaviors and the sub-components of the free energy are analyzed. It is hard to catch up why the linear and inversely linear behaviors come out in the two-different thickness regimes. As the conventional Kittel's law can be understood as a consequence of the competition between bulk energy and wall energy, more simplified model equations might be proposed based on the quantitative information obtained from the phase field model.

Reply to comment (R1.1): We thank the reviewer again for these constructive comments and suggestions. We have further designed a new phase field model to study the period-thickness relationship of skyrmions. In this method, a square skyrmion lattice model with varying in-plane size, or the period, was considered. The total energy densities of models with different periods and thicknesses are calculated to determine the optimal period for each thickness, as shown in Figure R1.1. For each STO/PTO/STO trilayer, a minimum in the energy density-period curve exists. We have used a series of polynomials to fit these curves to obtain the optimal periods and found the cubic polynomial could fit all these curves well. Figure R1.2 gives the variance of the obtained optimal period with the film thickness. It is found that the optimal period increases with the film thickness in the thickness range from 2 nm to 12 nm.

Figure R1.1 The total energy density curves for the STO/PTO/STO trilayers with different thicknesses for the skyrmion lattice models. The lowest total energy density for each thickness was chosen as the reference. The solid curves and dashed drop lines are guides to the eye. For better visibility, the curves of 2.0 nm and 1.6 nm were shifted downward by 1×10^5 and 2×10^5 J/m³, respectively.

Figure R1.2 The period-thickness relationship obtained by the phase-field simulations for the skyrmion lattice model.

To further understand the period-thickness relationship, the energy components for several thicknesses were analyzed. Figure R1.3a gives the evolution of different energy components with the period for the 12 nm film. It is found that the energy components are all monotonous around the optimized period: The bulk and electrostatic energy densities increase with the period, while the gradient and elastic ones decrease. In the classical Kittel model, the equations of energy components were deduced as the functions of film thickness and domain width (D) for the flux-closure and flux-open magnetic domain structures and the classical square root law was obtained from the competition of two energy groups: The one proportional to D (the anisotropic and magnetic energies, similar to the bulk and electrostatic energies here) and the one

inversely proportional to D (the domain wall energy, similar to the gradient energy here). In our studied case, the competition of the sum of the bulk and electrostatic energy densities and the sum of the gradient and elastic energy densities determines the optimal period.

It should also be noted that the bulk energy density of the 12 nm film would experience a minimum with the decrease of period ($d = 13$ nm). As the decrease of the film thickness, the position of the bulk energy minimum gradually shifts to the optimal period, as shown in Figure R1.3b,c. The gradient energy density curve shows a maximum when the thickness is 3.2 nm, as shown in Figure R1.3c. The position of this maximum also shifts to the optimal period when the film thickness further decreases to 1.6 nm, as shown in Figure R1.3d. Comparing the four panels in Figure R1.3, it is found that when $h \geq 4$ nm, the energy components are all monotonous near the optimal period. When $h < 4$ nm, the bulk and gradient energy densities show a minimum and a maximum close to the optimal period, respectively. As narrated in the last paragraph, the monotonicity of different energy components are the prerequisite for the deduction of the Kittel's law. We can now infer the bottom limit of the Kittel's law as $h = 4$ nm in this system. Thus, here we have summarized all thickness ranges for PTO layers where we believed that the period-thickness relationship in ultrathin films does not accord with Kittel's law.

Figure R1.3 The variance of energy components of the skyrmion lattice for the STO/PTO/STO trilayer with different PTO layer thicknesses. **a** 12 nm; **b** 4 nm; **c** 3.2 nm; **d** 1.6 nm. The energy components at the optimal periods were taken as the references.

Changes (R1.1): We have rewritten several paragraphs from Page 5 to Page 6 in the revised manuscript as marked by the red color and added Supplementary Fig. 7 in the revised manuscript.

Comment (R1.2): The planar view DF images (e.g., Fig. 3a) are difficult to identify the detailed structure. Authors argue the topological number is 1 as presented in Fig. 4b. It is not clear which area they calculate over. As an example, it is necessary to plot an enlarged view of planar dipole distribution and the corresponding skyrmion density map.

Reply to comment (R1.2): We appreciate the meaningful comments by the reviewer. Indeed, as mentioned by the reviewer, it is not very clear to identify the detailed structure in Fig. 3a. We then have replaced Fig. 3a with Figure R1.4 below showing more details.

Figure R1.4 Planar-view DF under-focus STEM imaging of $(\text{PTO}_4/\text{STO}_4)_{10}$ superlattices, showing the widespread occurrence of rounded polar skyrmions. The individual skyrmion is marked by white dotted circle.

The topological number is calculated based on the simulation results. Figure R1.5 gives the polarization distribution at the PTO/STO interface and the corresponding topological density distribution. It is found that the polarization vectors form the convergent pattern and the topological density mainly concentrates at the periphery of each skyrmion. By integrating the topological density, it is found that the topological charges of these skyrmions are all -1 . We corrected the expression “as confirmed by the experiment” to “as confirmed by the simulation results” in the revised manuscript.

Figure R1.5 The distributions of polarization **a** and topological density **b** at the

PTO/STO interface for the STO/PTO/STO model with the PTO layer thickness of 4 nm obtained from phase-field simulations.

In addition, based on the suggestions from reviewer 2, we also added some planar-view polar maps, as shown in Supplementary Figs. 15 on Page 17 in the revised supplementary materials.

Changes (R1.2): We have replaced Fig. 3a with Figure R1.4 on Page 14 and replaced Fig. 4a,b with Figure R1.5a,b on page 15 in the revised manuscript. We also added several sentences on Page 9 in the revised manuscript. In addition, we added some planar-view polar maps, as shown in Supplementary Figs. 15 on Page 17 in revised supplementary materials.

Fig. 3 | Characterization of the limit polar skyrmion. a Planar-view DF under-focus STEM imaging of $(\text{PTO}_4/\text{STO}_4)_{10}$ superlattices, showing the widespread occurrence of rounded polar skyrmions. The individual skyrmion is marked by white dotted circle. The 3D skyrmion maps are located under the under-focus STEM images.....

Fig. 4 | The topological number and possible formation mechanism of ferroelectric skyrmions. The distributions of polarization a and topological density b at the

PTO/STO interface for the STO/PTO/STO model with the PTO layer thickness of 4 nm obtained from phase-field simulations. **c** Dependence of the topological number (N_Q) on thickness. **d** RSM around (103) Bragg spot of the STO substrate for the (PTO₇/LSMO₇)₁₀ superlattices. **e** UV absorption measurements on the (PTO_{*n*}/STO_{*n*})₁₀ (*n* = 12, 4 u.c.) superlattices. A bare STO and single PTO film are also measured here for comparison. **f** Coercive voltage distribution map corresponding to different superlattice thicknesses.

(on Page 9 in revised manuscript).....The topological number was calculated based on the simulation results. Fig. 4a,b give the polarization distribution at the PTO/STO interface and the corresponding topological density distribution. It is found that the polarization vectors form a convergent pattern and the topological density mainly concentrates at the periphery of each skyrmion.By integrating the topological density, it is found that the topological charges of these skyrmions are all -1 . The topological charge of a skyrmion is related with the polarization direction of the skyrmion center: If the polar vector points to the $+z$ direction, the topological charge is $+1$, and if it points to the $-z$ direction, the topological charge is -1 .

We adjusted the order of several diagrams in Fig. 4. We also added the initial Fig. 4d,e,g,h,i in supplementary Fig. 22 on Page 20 in the revised supplementary materials.

Supplementary Fig. 22. PFM measurement. **a-h** PFM amplitude–voltage butterfly loop and phase–voltage hysteresis loop of a series of $(\text{PTO}_n/\text{STO}_n)_{10}$ ($n = 37, 28, 23, 19, 12, 9, 4, 2$ u.c.) superlattices. **i-k** Domain writing of the $(\text{PTO}_n/\text{STO}_n)_{10}$ ($n = 19, 12$ u.c.) superlattices.

Comment (R1.3): How can we understand the ultrathin thicknesses result in topologically non-trivial textures in (001) plane different from the thicker ones?

Reply to comment (R1.3): We thank the reviewer for this constructive comment. Let us discuss the possible skyrmions formation mechanism, which may help to understand the ultrathin thicknesses result in topologically non-trivial textures in (001) plane different from the thicker ones.

Based on Supplementary Table 2, the lattice mismatch between STO and PTO is almost zero. In order to screen the depolarization field, a large number of 180° domains are formed in the epitaxial single PTO films directly grown on STO substrates. When the PTO/STO superlattice is grown on the STO substrate, the depolarization field at the PTO/STO interface may govern the formation of skyrmions. So, skyrmions may behave like the 180° domains, with the ferroelectric rather than ferroelastic properties. When the single PTO/STO superlattice thickness is thicker, the effect of depolarization field on the whole PTO film is limited. The strip 180° domain was not completely cut by the depolarized field, and some nanometer-size elongated features appeared. But, near the PTO/STO interface, skyrmions characteristics is still obvious. When the single PTO/STO superlattice thickness is thinner, a large depolarization field at the PTO/STO interface becomes dominant in ultrathin layers. The depolarization field cuts the superlattice into the nanometer-size round skyrmions with about 6 nm. Thus, the depolarization field in the ultrathin thicknesses results in topologically non-trivial textures in (001) plane.

Skyrmions in thicker PTO/STO superlattices have elongated features. Skyrmions in the thinner superlattice feature with nanometer-size round characters.

Comment (R1.4): Many of the macroscopic characteristics in Fig. 4 seem to be not directly related to the main theme of the paper.

Reply to comment (R1.4): We thank the reviewer for this constructive comment. As suggested by the reviewer, we have further conducted detailed analysis to clarify the internal logic of our work.

Here, the ferroelectric layers are sandwiched by insulators which may result in a large depolarization field at the film interface, which will become even dominant in ultrathin layers and affect classic Kittel's law. So the absence of skyrmion satellite peaks in the PTO/LSMO superlattices may be associated with charge screening at the interface which may alter the depolarization fields signifying the importance of electrostatic boundary conditions. In addition, in order to explore the relationship between the ferroelectric skyrmion structure and the underlying functional properties, we have completed preliminary exploration of UV absorption behavior. The superlattices containing skyrmions show more intensive absorption over the whole visible range of wavelengths investigated. This may be derived from the complex hedgehog-like and Block-like polarization component in skyrmions PTO. Finally, the

PFM test for these PTO/STO superlattices further suggested that the ultrathin superlattice has specific ferroelectric and polarization reversal properties, which may be illuminative for the ferroelectric skyrmion-based device design and applications in the future.

Changes (R1.4): By combining the comment R1.2 with comment R1.4, We added initial Fig. 4d,e,g,h,i to supplementary Fig. 22 on Page 20 in the revised supplementary materials. We adjusted the order of several diagrams in Fig. 4. We also added related analysis on Page 10 to further clarify the relevance to the topic of this work in the revised manuscript.

(on Page 10 in revised manuscript).....**On the contrary, the ferroelectric layers sandwiched by insulators may result in a large depolarization field at the film interface, which will become even dominant in ultrathin layers and affect classic Kittel's law.....**

Reply to reviewers' questions and comments:

Ref number: NCOMMS-22-42842-T

Title: Absence of critical thickness for polar skyrmions with breaking the Kittel's law

Authors: Feng-Hui Gong, Yun-Long Tang, Yu-Jia Wang, Yu-Ting Chen, Bo Wu, Li-Xin Yang, Yin-Lian Zhu & Xiu-Liang Ma

20 February, 2023

Reply to reviewer 2 (R2):

We appreciate the positive comments by the reviewer that “the manuscript reports an interesting phenomenon that the skyrmion periods and superlattice thicknesses disobey Kittel's law in PTO/STO superlattices with ultrathin PTO layers”, “the data present in the manuscript are excellent and convincing”, and that “the theoretical simulation also gives a reasonable explanation of the experimental phenomenon.”. We also thank the reviewer for the positive evaluation that “I believe this discovery in ferroelectric topological domains is essential to research aiming at ferroelectric ultra-high density storage”, and that “I am sure this finding has great potential merits to be published in Nature Communication.”. In the meantime, the reviewer also raises some questions. We fully understand the reviewer's concerns, and here we will try to address all the questions and discuss all the comments one-by-one in the following. The revisions were indicated in **RED** characters in the revised manuscript.

Comment (R2.1): The polar skyrmion is a particle-like polar topology. Can “in-plane modulation periods” in RSM of cross-sectional samples fully represent the d (the periods of polar domain) in Kittel's law? Some direct evidence from the planar view might be considered, which helps to increase credibility. For example, adding some planar view STEM images and corresponding density statistics of polar skyrmions in typical systems, PTO_{*n*}/STO_{*n*} superlattices. We do directly see the density distribution of the skyrmions get increase as the PTO gets thinner, in Supplementary Fig 2. But, is it directly related to the periods of polar domain?

Reply to comment (R2.1): We thank the reviewer for this in-depth comment and suggestion. Indeed, as the reviewer mentioned, the polar skyrmion is a particle-like polar topology. We then have done further planar view STEM experiments based on the suggestions of the reviewer. We have the following considerations.

i: Generally, the modulation of satellite peaks in diffractions could represent the period of polar domain, as reported in previous skyrmion and vortex work^{18,37}. The d in Kittel's law also represents the period or width of polar domains, as listed in Supplementary Table 1.

ii: The obtained period by RSM is the average period information rather than diameter of polar domains, which is representative. And we also get the selected electron diffraction (SAED) from TEM cross-sectional samples (Supplementary Fig. 6). The SAED reflects the local period. The obtained period by RSM is consistent with that obtained period by SAED.

Next, we have done more STEM imaging to see more details, as suggested by the reviewer. STEM images of polar skyrmion in the $(\text{PTO}_n/\text{STO}_n)_{10}$ ($n = 37, 28, 23, 19, 12, 4$ u.c.) superlattices (Figure R2.1) were obtained. We fitted the relationship between the skyrmion density and superlattice thickness (Figure R2.2). With the decrease of superlattice thicknesses, the skyrmion density increases gradually. Here, the relationship between skyrmion density and single superlattice thickness is approximately inverse proportional function.

Figure R2.1 Under-focus STEM imaging of polar skyrmion of the $(\text{PTO}_n/\text{STO}_n)_{10}$ ($n = 37, 28, 23, 19, 12, 4$ u.c.) superlattices.

Figure R2.2 The relationship between the skyrmion density and superlattice thickness.

Changes (R2.1): We added Figure R2.1a,b,d to supplementary Fig. 11 on Page 15 in the revised supplementary materials.

Comment (R2.2): I am wondering if the topological domains in superlattice with thicker PTO layers are still kept as polar skyrmions? Maybe due to the image grayscale, it is hard to see obvious circular or elongated features in Extended Data Fig. 7 and Supplementary Fig. 2a. I suggest the author should add a mark on the polar domains for evaluating the size. The data of the cross-sectional view is very clear and detailed; If adding some data of the planar view (polar map or PFM), which will further prove that they are the larger skyrmions rather than other topologies.

Reply to comment (R2.2): We thank the reviewer for this in-depth comment and suggestion. Based on the reviewer's questions, we grew thicker $(\text{PTO}_{50}/\text{STO}_{50})_1$ trilayer (Figure R2.3a,b). The PTO layer is about 21 nm in thickness. Intensity modulations suggest the formation of polar skyrmions, which is similar to other thickness superlattices.

We added marks (white dotted circle or ellipse) on the polar domains for evaluating the size in Fig. 3a on Page 14 in revised manuscript and supplementary Figs. 9a, 10a, 11, 15, 16 on Pages 13-15, 17 in revised supplementary materials.

We also added some planar-view polar maps, which further proved that they are the larger skyrmions rather than other topologies. Figure R2.3c,d are the planar-view HAADF-STEM images of $(\text{PTO}_n/\text{STO}_n)_1$ ($n = 23, 12$ u.c.) trilayers. We extracted the polarization distribution of the white dotted areas, which clearly showed the skyrmion polar characteristic.

Figure R2.3 The ferroelectric skyrmions in $(\text{PTO}_n/\text{STO}_n)_1$ ($n = 50, 23, 12$ u.c.) trilayers. **a, b** The cross-sectional DF image and low-magnification HAADF-STEM image of $(\text{PTO}_{50}/\text{STO}_{50})_1$ trilayer. **c, d** The planar-view HAADF-STEM images and extracted skyrmion polarization mappings of $(\text{PTO}_n/\text{STO}_n)_1$ ($n = 23, 12$ u.c.) trilayers. The white dotted circles and ellipse mark skyrmions areas.

Changes (R2.2): We added Figure R2.3 to supplementary Fig. 15 on Page 17 in revised supplementary materials and added some sentences on Pages 7 and 8 in revised manuscript.

(on Pages 7 and 8 in revised manuscript).....**Supplementary Fig. 15a-d** are the DF image of $(\text{PTO}_{50}/\text{STO}_{50})_1$ trilayer and planar-view HAADF-STEM images of $(\text{PTO}_n/\text{STO}_n)_1$ ($n = 23, 12$ u.c.) trilayers.....The polarization distributions of the white dotted areas, clearly show the skyrmion polar characteristic (Supplementary Fig. 15).

The planar-view polar map further proves that they are the skyrmions rather than other topologies (Supplementary Fig. 15).

We also added marks (white dotted circle or ellipse) on the polar domains for evaluating the size and one sentence (The individual skyrmion is marked by white dotted circle or ellipse) in Fig. 3a on Page 14 in revised manuscript and supplementary Figs. 9a, 10a, 11, 15, 16 on Pages 13-15, 17 in revised supplementary materials.

Comment (R2.3): In Fig 3c, does the polar topology constructed in both STO and PTO layers? In addition, the cross-section view has the feature of a vortex structure, is there any possible a new polar type of topology?

Reply to comment (R2.3): We thank the reviewer for this in-depth and stimulating discussion, which is illuminative for studying the polar structures here. Indeed, as noticed by the reviewer, in previous works, when the STO layer is about a few unit cells, the PTO polarization could drive a polar (even some special polar structures) STO layer³⁰. In fact, in the thicker PTO/STO superlattices, 1 ~ 2 STO unit cells near the PTO/STO interface could probably become polar. In our work, in the ultrathin (PTO₂/STO₂)₁₀ superlattices grown on the STO substrate, there is also polarization phenomenon (ion displacement) in some regions of STO layer due to the penetration of PTO polarization. Therefore, in the ultrathin (PTO₂/STO₂)₁₀ superlattices, a portion of the polar topology may indeed exist in the STO layer.

Changes (R2.3): We further added marks and detailed instructions in supplementary Fig. 14 on Page 16 in the revised supplementary materials. We added some sentences on Page 7 in the revised manuscript. The new reference 30 was also added on Page 21 in the revised manuscript.

(on Page 7 in the revised manuscript).....Furthermore, for the (PTO₂/STO₂)₁₀ superlattices, polarization map (Fig. 3c) based on the HAADF-STEM images (Fig. 3b) also indicates the signal of the polar skyrmions, which corresponds to the red dotted box in Supplementary Fig. 14. Fig. 3d indicates that the ion displacement of closed ring along the out-of-plane direction is about 0 ~ 20 pm.

30. Abid, A. Y. et al. Creating polar antivortex in PbTiO₃/SrTiO₃ superlattice. *Nat. Commun.* **12**, 2054 (2021).

Comment (R2.4): I am personally interested in the calculation of topological numbers in Figure 4b. Are these calculated by simulation results in Supplementary Fig. 5, or by the polar maps obtained from experiments? What exactly does the “as confirmed by the experiment” mean?

Reply to comment (R2.4): We appreciate the meaningful comments by the reviewer. The topological number (revised Fig. 4c) is calculated based on the simulation results. We corrected the inappropriate expression “as confirmed by the experiment” to “as confirmed by the simulation results” on page 9 in the revised manuscript.

Figure R2.4a,b here gives the polarization distribution at the PTO/STO interface and the corresponding topological density distribution. It is found that the polarization vectors form the convergent pattern and the topological density mainly concentrates at the periphery of each skyrmion. By integrating the topological density, it is found that

the topological charges of these skyrmions are all -1 .

Figure R2.4 The distributions of polarization **a** and topological density **b** at the PTO/STO interface for the STO/PTO/STO model with the PTO layer thickness of 4 nm.

Changes (R2.4): We corrected the expression “as confirmed by the experiment” to “as confirmed by the simulation results” on Page 9 in the revised manuscript. We added Figure R2.4 to Fig. 4 on Page 15 in the revised manuscript. We adjusted the order of several diagrams in Fig. 4. We added some sentences on Page 9 in the revised manuscript.

Fig. 4 | The topological number and possible formation mechanism of ferroelectric skyrmions. The distributions of polarization **a** and topological density **b** at the PTO/STO interface for the STO/PTO/STO model with the PTO layer thickness of 4 nm obtained from phase-field simulations. **c** Dependence of the topological number (N_Q) on thickness. **d** RSM around (103) Bragg spot of the STO substrate for the $(\text{PTO}_7/\text{LSMO}_7)_{10}$ superlattices. **e** UV absorption measurements on the $(\text{PTO}_n/\text{STO}_n)_{10}$ ($n = 12, 4$ u.c.) superlattices. A bare STO and single PTO film are also measured here for comparison. **f** Coercive voltage distribution map corresponding to different superlattice thicknesses.

(on Page 9 in revised manuscript).....We proved that skyrmions can exist in ultrathin superlattices and explored the evolution behavior of skyrmions with thickness. The topological number was calculated based on the simulation results. Fig. 4a,b give the polarization distribution at the PTO/STO interface and the corresponding topological density distribution. It is found that the polarization vectors form a convergent pattern and the topological density mainly concentrates at the periphery of each skyrmion.....By integrating the topological density, it is found that the topological charges of these skyrmions are all -1 . The topological charge of a skyrmion is related with the polarization direction of the skyrmion center: If the polar vector points to the $+z$ direction, the topological charge is $+1$, and if it points to the $-z$ direction, the topological charge is -1 . As the thickness decreases.....

Reply to reviewers' questions and comments:

Ref number: NCOMMS-22-42842-T

Title: Absence of critical thickness for polar skyrmions with breaking the Kittel's law

Authors: Feng-Hui Gong, Yun-Long Tang, Yu-Jia Wang, Yu-Ting Chen, Bo Wu, Li-Xin Yang, Yin-Lian Zhu & Xiu-Liang Ma

20 February, 2023

Reply to reviewer 3 (R3):

First, we thank the reviewer for the meaningful discussions on the potential applications and research progresses in the field of ferroelectric films that “ultrathin ferroelectric devices are important for high-density data storage applications and other nano-electronic technologies”, “however, the depolarization field and interface effects in thin films may become dominant at reduced thicknesses which significantly affect the domain structure and switching properties of ferroelectric materials”, and that “the availability of advanced thin film growth and structural imaging technology has boosted the research on dimensionally confined ferroelectrics which can play an essential role in developing a fundamental understanding of non-conventional behavior observed at reduced dimensions in ferroelectrics.”.

Second, we appreciate the positive summary by the reviewer that “gong *et al.* have studied the effect of layer thickness on the domain structure in $\text{PbTiO}_3/\text{SrTiO}_3$ superlattices down to two unit cells (u.c.)”, “by using X-ray diffraction and scanning transmission electron microscopy, the authors confirmed the formation of polar skyrmions in the heterostructures which could be retained at an ultralow thickness of the PbTiO_3 layer (2 u.c.)”, “the authors particularly observed the deviation from Kittel's law (which relates the domain width or domain period with the square root of film thickness) at these reduced dimensions”, “this deviation or ‘breaking’ from Kittel's law is supported and explained based on phase-field calculations”, and that “polarization switching and the effect of electric boundary conditions have also been studied in ultrathin layers.”.

At the same time, the reviewer also raised some concerns about the further clarifying of novelty and the discussion depth. Here we will try to address all the questions and discuss the comments one-by-one in the following. We thank the useful references recommended by the reviewer, which do help to explain the meaning and novelty of our work. Corresponding revisions are highlighted in **RED** in the marked, revised manuscript.

Comment (R3.1): The novelty of this work is unclear. The breaking from Kittel's law at ultralow thicknesses has been theoretically predicted and experimentally shown in previous works (J. Appl. Phys., 1993, 74(10), 6012-6019 also ref. 5, Ferroelectrics, 1997, 202(1), 267-274 also ref. 8 & J. Appl. Phys, 2008, 104(6), 064109).

Reply to comment (R3.1): We appreciate the meaningful comments by the reviewer, and we thank the references recommended here, which is important for clarifying the

novelty of our work. We also note that most of these references were cited in our work and the left reference recommended here was added as Ref. 13 in the revised manuscript (listed at the end of **R3.1** here). Next, we will further clarify the novelty of our work. We have the following considerations.

i: The two theoretical works suggested by the reviewer (we also have cited in our previous version) indeed deduced the breaking of Kittel's law based on elastic energy (J. Appl. Phys., 1993, 74(10), 6012-6019 also ref. 5) and electrostatic interactions (Ferroelectrics, 1997, 202(1), 267-274 also ref. 9), where a/c and a_1/a_2 domain distributions were studied. While these works are illuminative, polar topologies, such as the skyrmions studied in our work (which exist in ultrathin PTO layers and were not involved in these pervious works) and their distributions in ultrathin PTO layers, were not revealed in these theoretical works. Differently, we have combined both experiments and phase-field calculations to clarify the unusual breaking from Kittel's law of these polar skyrmions. Thus we believe our work displays novel realizations that should arouse the curiosity of ferroelectric community.

ii: Experimentally, as mentioned by the reviewer (J. Appl. Phys, 2008, 104(6), 064109), the relationship between period and thickness of c/c domain was studied here. They have revealed that for the PTO films with thinner thickness range (6-100 unit cells) their stripe width was found to be proportional to the square root of the film thickness, which is completely different from our work (as shown in Figure R3.1).

Figure R3.1 **a, b** The relationship between periods and thicknesses of PbTiO_3 layers containing skyrmions. **b** If we ignore the difference between period 6.8, 6.7, 5.2, 6.8 nm, the period is constant as the superlattice thickness decreases.

iii: Ferroelectric skyrmion has potential applications in high-density storage memories. In the post-Moore era, the size limit problem must be considered in the design and manufacture of electronic devices. Therefore, it is important to clear the critical thickness issues of polar skyrmions for potential applications. In our work, by reducing the single PTO/STO layer thickness, we found that polar skyrmions exist not only in ultrathin $(\text{PTO}_2/\text{STO}_2)_{10}$ superlattices, but also in $(\text{PTO}_2/\text{STO}_2)_1$ trilayers, which will be important for the future study of skyrmion-based ferroelectric physics. In addition, the evolution of these polar structures at high temperatures was highlighted in our work. We believe these aspects are also the new and important realizations on polar

skyrmions.

13. Takahashi, R., Dahl, Ø., Eberg, E., Grepstad, J. K. & Tybell, T. Ferroelectric stripe domains in PbTiO_3 thin films: Depolarization field and domain randomness. *J. Appl. Phys.* **104**, 064109 (2008).

Comment (R3.2): The type of medium immediate to the ferroelectric layer (vacuum/metal/insulator) plays an important role in determining the electrostatic boundary conditions. In the current work, the ferroelectric layers are sandwiched between insulators which may result in a large depolarization field at the film interface that becomes dominant in ultrathin layers and affect the breakdown of Kittel's law. Previously developed models for the sub-Kittel regimes provide insight into the electrostatic description of ultrathin ferroelectric layers in different settings (Ferroelectrics, 1997, 202(1), 267-274 & R. Soc. Open Sci., 2020, 7(11), 201270). In the current work, the relationship between film thickness and the period of polar skyrmions has only been empirically established while the discussion based on phase-field simulation results seems rudimentary. It is suggested to provide further discussion taking into account the electrostatic boundary conditions, especially the depolarization field, and provide insight into the correlation between layer thickness and domain modulation. The fitting expression and parameters used in Fig. 2a need to be further analyzed keeping in view the previous models for sub-Kittel regions.

Reply to comment (R3.2): We thank the reviewer for this constructive discussion and the very meaningful and useful references. Inspired by the discussion and references, we further adopted a new method to study the period-thickness relationship by phase-field simulations. In this method, a square skyrmion lattice model with varying in-plane size, or the period, was considered. The total energy densities of models with different periods and thicknesses were calculated to determine the optimal period for each thickness, as shown in Figure R3.2. For each STO/PTO/STO trilayer, a minimum in the energy density-period curve exists. We have used a series of polynomials to fit these curves to obtain the optimal periods and found the cubic polynomial could fit all these curves well. Figure R3.3 gives the variance of the obtained optimal period with the film thickness. It is found that the optimal period increases with the film thickness in the thickness range from 2 nm to 12 nm.

Figure R3.2 The total energy density curves for the STO/PTO/STO trilayers with different thicknesses for the skyrmion lattice model. The lowest total energy density for each thickness was chosen as the reference. The solid curves and dashed drop lines are guides to the eye. For better visibility, the curves of 2.0 nm and 1.6 nm were shifted downward by 1×10^5 and 2×10^5 J/m³, respectively.

Figure R3.3 The period-thickness relationship obtained by the phase-field simulations for the skyrmion lattice model.

To further understand the period-thickness relationship, the energy components for several thicknesses were analyzed. Figure R3.4a gives the evolution of different energy components with the period for the 12 nm film. It is found that the energy components are all monotonous around the optimized period: The bulk and electrostatic energy densities increase with the period, while the gradient and elastic ones decrease. In the classical Kittel model, the equations of energy components were deduced as the functions of film thickness and domain width (D) for the flux-closure and flux-open magnetic domain structures and the classical square root law was obtained from the competition of two energy groups: the one proportional to D (the anisotropic and magnetic energies, similar to the bulk and electrostatic energies here) and the one inversely proportional to D (the domain wall energy, similar to the gradient energy here). In our studied case, the competition of the sum of the bulk and electrostatic energy densities and the sum of the gradient and elastic energy densities determines the optimal period.

It should also be noted that the bulk energy density of the 12 nm film would experience a minimum with the decrease of period (located at about $d = 13$ nm). As the decrease of the film thickness, the position of the bulk energy minimum gradually shifts to the optimal period, as shown in Figure R3.4b and R3.4c. The gradient energy density curve shows a maximum when the thickness is 3.2 nm, as shown in Figure R3.4c. The position of this maximum also shifts to the optimal period when the film thickness further decreases to 1.6 nm, as shown in Figure R3.4d. Comparing the four panels in Figure R3.4, it is found that when $h \geq 4$ nm, the energy components are all monotonous near the optimal period. When $h < 4$ nm, the bulk and gradient energy densities show a

minimum and a maximum close to the optimal period, respectively. As narrated in the last paragraph, the monotonicity of different energy components are the prerequisite for the deduction of the Kittel's law. We can now infer the bottom limit of the Kittel's law is $h = 4$ nm in this system. Thus, here we have summarized all thickness ranges for PTO layers where we believed that the period-thickness relationship in ultrathin films does not accord with Kittel's law.

Figure R3.4 The variance of energy components of the skyrmion lattice for the STO/PTO/STO trilayer with different PTO layer thicknesses. **a** 12 nm; **b** 4 nm; **c** 3.2 nm; **d** 1.6 nm. The energy components at the optimal periods were taken as the references.

As for the electrostatic boundary condition and the depolarization field, it is well known that the effect of depolarization field should be more important for thinner films. Figure R3.5 shows the electric field and polarization profiles along the film normal across the center of the skyrmion. It is found that the polarization profiles are generally the same for different film thicknesses and the electric field profiles are obviously different for thicker and thinner films. When the film thickness is 20 nm, the electric field in the PTO layer, or the depolarization field, mainly concentrates near the PTO/STO interfaces and gradually decreases away from the interfaces. In the middle of the PTO layer, the depolarization field is nearly zero, as shown in Figure R3.5a. When the film thickness decreases, the depolarization field in the middle of the PTO layer gradually increases, as shown in Figure R3.5b-d. In other words, the effect of depolarization field becomes more important for thinner films.

Figure R3.5 The electric field and polarization profiles along the film normal for the STO/PTO/STO trilayers with different thicknesses. **a** 20 nm; **b** 8 nm; **c** 4 nm; **d** 1.6 nm.

The reviewer's discussion is very important and enlightening, improving the logic and rigor of this work. We thank again the reviewer for this constructive discussion and the very meaningful and useful references.

Changes (R3.2): We have rewritten several paragraphs from Page 5 to Page 6 in the revised manuscript as marked by the red color and added Supplementary Figs. 7,8 on Page 12 in the revised supplementary materials. The two literatures are added as Ref. 9 and 60 in the revised manuscript.

Comment (R3.3): How do the authors define critical thickness here? Is it the layer thickness below which the domain period breaks from Kittel's law or below which the ferroelectricity disappears?

Reply to comment (R3.3): We thank the reviewer for this meaningful comment. The definition of critical thickness here is "the critical PTO layer thickness could hold the formation of polar skyrmions and their statistical distribution features.", which can be found on Page 3 in the manuscript.

Comment (R3.4): Based on HAADF and EDS images in Extended Fig 1 and 2 respectively, both the PbTiO_3 and SrTiO_3 layers in $(\text{PTO}_2/\text{STO}_2)_{10}$ superlattice seem to be 3 u.c. thick rather than 2 u.c. Can the authors explain how this superlattice is determined to be $(\text{PTO}_2/\text{STO}_2)_{10}$?

Reply to comment (R3.4): We thank the reviewer for this valuable comment and we will further explain and refine. i: Based on previous work¹⁸, we believed that 2 PTO unit cells may contain three Pb atomic layers, and 2 STO unit cells may contain three

Sr atomic layers. The interface unit cell composed of Pb-Sr belong to neither PTO unit cell nor STO unit cell, so the n in $(\text{PTO}_n/\text{STO}_n)_{10}$ superlattices does not contain Pb-Sr interface unit cell, as shown in Figure R3.6. Previous work for n in $(\text{PTO}_n/\text{STO}_n)_{10}$ superlattices also did not include Pb-Sr interface cells¹⁸. ii: Our aim is to grow high quality $(\text{PTO}_2/\text{STO}_2)_{10}$ superlattices. We have grown a large number of ultrathin superlattices by manual and automatic control. Up to now, because of composition fluctuations and atomic diffusion, we think that the PLD technology cannot precisely achieve a true atomic scale flat interface, especially when the n is less than 3 unit cells. We replaced the original supplementary Fig. 2h.

Figure R3.6 Definition of superlattice thickness. **a** Schematic diagram of $(\text{PTO}_2/\text{STO}_2)_{10}$ superlattice. **b** HAADF-STEM image of $(\text{PTO}_2/\text{STO}_2)_{10}$ superlattices.

Changes (R3.4): We replaced the original supplementary Fig. 2h. We also added Figure R3.6 to supplementary Fig. 2h on Page 8 in revised supplementary materials.

Comment (R3.5): Compared to $(\text{PTO}_{12}/\text{STO}_{12})_{10}$ superlattice, the evolution of domain structure in $(\text{PTO}_2/\text{STO}_2)_{10}$ superlattice at high temperatures would be more interesting since this may provide some insight into the order parameters driving the stability of skyrmions in ultrathin layers.

Reply to comment (R3.5): We thank the reviewer for this valuable comment and suggestion about the stability of skyrmions in ultrathin layers. We systematically studied the evolution of skyrmions with temperature for different thicknesses by phase field calculation. As shown in Figure R3.7, thinner films possess smaller Curie temperatures. It is reasonable since the polarization magnitude of a thinner film tends to be smaller.

To gain more insight into the effect of temperature on the stability of skyrmions, we calculated the evolution of energy components with temperature for different thicknesses, as shown in Figure R3.8. It is found that with the increase of temperature, the bulk, gradient and electrostatic energy densities gradually decrease to zero and the elastic energy density convergent to some certain value, both for the case of ultrathin and thicker films. Thus, the temperature-driven instability of skyrmions for ultrathin and thicker films might have the same root.

Figure R3.7 **a** The evolution of averaged polarization magnitude in the PTO layer with temperature for the STO/PTO/STO trilayers with different thicknesses. **b** The variance of Curie temperature with the film thickness.

Figure R3.8 The evolution of energy components with temperature for the STO/PTO/STO trilayers with the thickness of 1.6 nm **a** and 12 nm **b**.

As suggested, we have also performed in-situ heating experiment for the $(\text{PTO}_2/\text{STO}_2)_{10}$ superlattices (Figure R3.9). In Figure R3.9a, the black dot contrast region should be the polar skyrmions, which disappeared around 423K (150°C).

Figure R3.9 In-situ heating of the $(\text{PTO}_2/\text{STO}_2)_{10}$ superlattices. **a-f** Temperature-dependent DF images from 323 K to 473 K.

Changes (R3.5): We have added several sentences on Page 9 in the revised manuscript and added Figure R3.7 as Supplementary Fig. 19 on Page 19 in the revised supplementary materials. We also added Figure R3.9 as Supplementary Fig. 18 on Page 18 in the revised supplementary materials.

(on Page 9 in revised manuscript)..... We also studied the evolution of skyrmions in other thickness superlattices with temperature by experiments and phase-field simulations (Supplementary Figs. 18,19). As shown in Supplementary Fig. 19, the

averaged polarization magnitude decreases with the increase of temperature for all PTO layers considered and thinner films possess smaller Curie temperatures. The calculated Curie temperature of the 4 nm PTO layer is about 710 K, close to the experimental result of the 4.8 nm PTO layer, 623 K.

Comment (R3.6): The effect of using LSMO as well as UV absorption behavior has been studied, however, it is not obvious how the obtained results support the main discussion or conclusion of the manuscript. For example, the absence of polar skyrmions in the PTO/LSMO superlattice may be associated with charge screening at the interface which may alter the depolarization fields signifying the importance of electrostatic boundary conditions.

Reply to comment (R3.6): We thank the reviewer again for this constructive comment and meaningful discussion. As discussed in above questions, we agree that the interfacial electrostatic boundary conditions are crucial for understanding the polar structures here. Here, when the STO layer is replaced by LSMO ($\text{La}_{0.7}\text{Sr}_{0.3}\text{MnO}_3$) electrode, the interfacial depolarized field will tend to be screened, thus the skyrmions disappear, as noticed by the reviewer. Differently, the ferroelectric layers sandwiched between insulators may result in a large depolarization field at the film interface which becomes dominant in ultrathin layers and affects the breakdown of Kittel's law. In ultrathin superlattices, the depolarization field may govern the entire PTO layer. In addition, in order to explore the connection between the ferroelectric skyrmion structure and the underlying functional properties, we have completed preliminary exploration of UV absorption behavior. The superlattices containing skyrmions show more intensive absorption over the whole visible range of wavelengths investigated. This may be derived from the complex hedgehog-like and Block-like polarization component in skyrmions PTO. As suggested by the reviewer, we agree that more detailed works are still needed in the future, so that we could get more information on the structure-properties of these skyrmions in ultrathin films.

Changes (R3.6): We added one sentence on Page 10 in revised manuscript.

(on Page 10 in revised manuscript).....**The absence of polar skyrmions in the PTO/LSMO superlattices may be associated with charge screening at the interface which may alter the depolarization fields signifying the importance of electrostatic boundary conditions.** On the contrary.....

Comment (R3.7): It is not very obvious how the out-of-plane PFM measurements were obtained without a conductive bottom electrode or proper grounding since the substrate is insulative. Authors may add more information regarding sample configuration for the PFM measurement.

Reply to comment (R3.7): We thank the reviewer for this constructive comment. As noticed by the reviewer, the conducting electrode is important for a more quantitative PFM experiment. According to the reviewer's suggestion, we added the sample configuration information in the revised manuscript. Before the PFM experiment, the conductive silver glue was applied to the iron sheet, the substrate side of the sample was glued to the iron sheet, and the sample was heated to 60 °C and baked for 20 min. Surface morphology, the domain writing, hysteresis loops and out-of-plane

amplitude/phase images were collected using the lithography PFM and the Dual AC Resonance Tracking (DART) mode at room-temperature (Cypher, Asylum Research). A DART model can relieve the topographic characteristics interference. Conductive Ti/Ir-coated silicon cantilevers (Asylum Research, ASYELEC-01-R2) were used for PFM measurements. The typical tip radius is about 7 nm and the force constant is $\sim 2 \text{ N m}^{-1}$.

Here, we learn from previously published works⁷⁰ that BaTiO₃ ultrathin films could be well studied without bottom electrodes, via PFM. We just apply voltage signal and get only voltage signal, which may be different from the measurement of an *I-V* curve, where requires the use of electrodes.

Changes (R3.7): We added the sample configuration information on Page 19 and above reference 70 on Page 23 in the revised manuscript.

(on Page 19 in revised manuscript).....**Before the PFM test, the conductive silver glue was applied to the iron sheet, the substrate side of the sample was glued to the iron sheet, and the sample was heated to 60 °C and baked for 20 min.** Surface morphology, the domain writing, hysteresis loops and out-of-plane amplitude/phase images were collected using the lithography PFM and the Dual AC Resonance Tracking (DART) mode at room-temperature (Cypher, Asylum Research).....

70. Dubourdieu, C. et al. Switching of ferroelectric polarization in epitaxial BaTiO₃ films on silicon without a conducting bottom electrode. *Nat. Nanotech.* **8, 748-754 (2013).**

Comment (R3.8): It is stated in line 325 “Our results indicate that the critical thickness for polar topologies may be even thinner than the maintaining of ferroelectricity in ultrathin ferroelectric films up to even nearly vanishing, which will be important for designing novel electronic devices in the post-Moore era, such as ultrahigh density memories”. Although the individual layer of PbTiO₃ in (PTO₂/STO₂)₁₀ superlattice is 2 u.c. thin, the overall thickness of the film is still >15 nm which is much thicker than other ferroelectric thin films reported in the literature with switchable polarization. In this regard, will the polar skyrmions and ferroelectricity sustain in a single 2 u.c. thin PbTiO₃ layer sandwiched between two SrTiO₃ layers? Based on this, can authors further elaborate what is the general applicability of the results obtained in this work and how these superlattices may help in designing ultrahigh-density memories or cite relevant literature?

Reply to comment (R3.8): We thank the reviewer for this in-depth comment and meaningful suggestion. Based on the reviewer's comments, we further deposited the (PTO₂/STO₆)₁ and (PTO₂/STO₂)₁ trilayers on the STO substrate, as shown in Figure R3.10a,b. The PTO/STO interface is sharp and the film is in good epitaxy with the substrate. Figure R3.10c is the cross-sectional HAADF-STEM image, which shows obvious Ti⁴⁺ displacement. It is consistent with the displacement ferroelectrics characteristics. Thus, based on the microstructure analysis, we believe that ferroelectricity exists in the single 2 u.c. PTO layers.

Figure R3.10d,e are the cross-sectional HAADF-STEM image and polarization map, respectively. Figure R3.10f is the planar-view HAADF-STEM image of the

($\text{PTO}_2/\text{STO}_2$)₁ trilayer. We further extracted the polarization map, as shown in Figure R3.10g, which are from the white dotted circles areas in f. The polarization distribution of the cross-sectional and the planar-view proves that the polar skyrmions are also maintained in single 2 u.c. PTO layers sandwiched between 2 u.c. STO layers. In addition, when the STO layer is about a few unit cells, the PTO polarization could drive a polar (even some special polar structures) STO layer³⁰. In fact, in the thicker PTO/STO superlattices, 1 ~ 2 STO unit cells near the PTO/STO interface could probably become polar. In our work, in the ultrathin ($\text{PTO}_2/\text{STO}_2$)₁ trilayer grown on the STO substrate, there is also polarization phenomenon (ion displacement) in some regions of STO layer due to the penetration of PTO polarization.

Therefore, based on a series of ($\text{PTO}_n/\text{STO}_n$)₁₀ and ($\text{PTO}_n/\text{STO}_n$)₁ (supplementary Figs. 15,16) experimental results, skyrmions exist not only in ultrathin superlattices, but also in ultrathin trilayers. Our conclusions may have general applicability.

Figure R3.10 The ferroelectricity and polar skyrmions in the ultrathin trilayers. **a, b** The HAADF-STEM images of ($\text{PTO}_2/\text{STO}_6$)₁ and ($\text{PTO}_2/\text{STO}_2$)₁ trilayers. **c** The HAADF-STEM image shows clear Ti^{4+} displacement. The yellow circle represents Pb atoms, the red circle represents Ti atoms. **d, e** The HAADF-STEM image and corresponding polarization map. **f** The planar-view HAADF-STEM images of ($\text{PTO}_2/\text{STO}_2$)₁ trilayer. **g** Polarization map extracted from the white dotted circles areas in **f**.

Finally, we cited the relevant literature for the skyrmions ultrahigh-density memories.

Changes (R3.8): We added Figure R3.10 to supplementary Fig. 16 on Page 17 in the revised supplementary materials. We added the above references on Page 21 in the revised manuscript. We also added some sentences on Page 8 and added two words on Page 11 in revised manuscript.

(on Page 8 in revised manuscript).....**Supplementary Fig. 16 are the cross-sectional and planar-view HAADF-STEM images of ($\text{PTO}_2/\text{STO}_2$)₁ trilayer. The polarization**

distribution of the white dotted areas, clearly show the skyrmion polar characteristic (Supplementary Fig. 16).....

(on Page 11 in revised manuscript).....Polar skyrmions can be maintained in ultrathin superlattices and trilayers with PTO layer as thin as two unit cells.....

10. Luo, S. J. & You, L. Skyrmion devices for memory and logic applications. *APL Mater.* **9**, 050901 (2021).

11. Tomasello, R. et al. A strategy for the design of skyrmion racetrack memories. *Sci. Rep.* **4**, 6784 (2014).

12. Kang, W., Huang, Y. Q., Zhang, X. C., Zhou, Y. & Zhao, W. S. Skyrmion-electronics: An overview and outlook. *Proceedings of the IEEE* **104**, 2040-2061 (2016).

We thank again the reviewer for the insightful questions and comments to improve our work. We fully understand the reviewer's concerns, and here we addressed all the questions and discussed the comments one-by-one. Corresponding revisions are highlighted in **RED** in the marked, revised manuscript.

Reply to reviewers' questions and comments:

Ref number: NCOMMS-22-42842-T

Title: Absence of critical thickness for polar skyrmions with breaking the Kittel's law

Authors: Feng-Hui Gong, Yun-Long Tang, Yu-Jia Wang, Yu-Ting Chen, Bo Wu, Li-Xin Yang, Yin-Lian Zhu & Xiu-Liang Ma

20 February, 2023

Reply to reviewer 4 (R4):

We appreciate the meaningful comments by the reviewer. The reviewer raises some specific questions and comments which can be summarized into four major aspects. We fully understand the reviewer's concerns, and here we will try to address all the questions and discuss all the comments one-by-one in the following. Corresponding revisions are highlighted in **RED** in the marked, revised manuscript. Overall, we have added more related analysis and experiments in the revised manuscript and supplementary materials.

Comment (R4.1): The first order satellite is often not sufficient for complex characterization since higher order ones contain more detail concerning the structure.

Reply to comment (R4.1): We thank the reviewer for this in-depth comment. Indeed, we agree that higher order satellite peaks should contain more detailed information. The 002 and 103 RSMs of the superlattices were then obtained and studied, as shown in Figure R4.1. By comparing the 002 and 103 diffraction peaks, we find that the extracted results from 002 and 103 diffraction patterns are consistent, and the obtained skyrmions periodicity is consistent.

Moreover, we have studied and obtained a variety of data by the powerful function of transmission electron microscopy (TEM) and STEM images up to the atomic level. These results could further confirm the existence of skyrmion in a series of superlattices. All the methods used in our works get the consistent results, which reveal the detailed evolution of skyrmions here.

Figure R4.1 RSM analysis. **a-d** RSMs of the (002) and (103) diffraction for $(\text{PTO}_n/\text{STO}_n)_{10}$ ($n = 37, 28$ u.c.) superlattices grown on STO substrates. Figure R4.1**b, d** are from Fig. 1**b, d** in the manuscript, respectively.

Changes (R4.1): We added one sentence on Page 16 in the Methods (High-resolution RSM) in the revised manuscript.

(on Page 16 in revised manuscript).....**The (103) diffraction spot contains both in-plane and out-of-plane information.**

Comment (R4.2): The large-scale twisting nature of the skyrmion leads to broad peaks in reciprocal space which is maybe the case for the 9/9 superlattice but the signal to noise is very poor. The other thicknesses show very different satellite patterns likely from planar ferroelectric phases (e.g. a_1/a_2), a/c order (12/12), and perhaps vortex phases. Regardless the satellite structures are evolving with thickness meaning that the real-space configuration is drastically changing. Kittel's scaling applies only in the case where the order is constant and is only changing in size. Likely these are mixed phase samples with complex heterogeneous order. As shown in previous work and the phase-field work (e.g. Small Methods 6, 2200486 (2022)), the phases are strongly dependent on the superlattice period and the strain state. This system is very unlikely to show the consistent order that can be used for a scaling study. Without a complex understanding of what phases are present and how they are mixed, one cannot try to apply a simple scaling argument.

Reply to comment (R4.2): We appreciate the reviewer's comments and discussions. Indeed, we agree that the phase components must be clarified first, so that the discussions on the Kittel's law will be reasonable. Next, we carried out detailed analysis and comprehensive understanding to eliminate reviewers' doubts and confirm the credibility of our results.

First, STO has a cubic crystal structure, $a = b = c = 3.905 \text{ \AA}$. PTO has a tetragonal crystal structure at room-temperature, $a = b = 3.899 \text{ \AA}$, $c = 4.153 \text{ \AA}$ (Supplementary

Fig. 12). The possible domain structures in PTO films under strains are shown in Figure R4.2. We agree that periodic a/c , a_1/a_2 or vortex domains in the PTO/STO superlattices may show similar RSM super-modulations, as also noticed by the reviewer. But this is not the case in our work. We have several considerations below:

Figure R4.2 **a** Schematic diagram of the c domain and a domain. **b** Schematic diagram of the c/c 180° domain, a/c 90° domain and a_1/a_2 domain.

i: If a_1/a_2 domains are present in the superlattices, the Bragg peaks and satellite peaks of a_1/a_2 domains should appear at the positions outlined with the black dashed lines in Figure R4.3a-h ($q_x = 7.7 \text{ nm}^{-1}$). The a_1 and a_2 domains are equivalent in-plane, so there will be two symmetrical diffraction spots in close proximity at Bragg peak. However, neither the a_1/a_2 Bragg peaks nor the a_1/a_2 satellite peaks were found in all RSM. Similarly, if periodic a/c domains are present in the superlattices, the Bragg diffraction spots of the c domains will appear at the positions outlined with the red dotted lines in Figure R4.3a-h ($q_x = 7.2 \text{ nm}^{-1}$). However, no Bragg diffraction spots of c domains can be identified along the red dotted lines. We then speculate that no a/c or a_1/a_2 domains form in $(\text{PTO}_n/\text{STO}_n)_{10}$ superlattices grown on the STO substrates. In addition, strain analysis shows that a/c domains and a_1/a_2 domains hardly appear in the superlattices grown on the STO substrates (Supplementary Table 2).

ii: a/c , a_1/a_2 , vortex and flux-closure are ferroelastic domains modulated by tensile strain, whereas c/c 180° domain and skyrmion are ferroelectric domains significantly affected by electric boundaries. Strain analysis helps us understand that a/c and a_1/a_2 domains unlikely exist in the PTO/STO superlattices grown on the STO substrates (Supplementary Table 2) since the lattice mismatch between STO and PTO is almost zero. Next, when the PTO/STO superlattices were grown on the scandate substrates, a periodic vortex array maybe formed in the PTO/STO superlattices as reported previously. However, due to the effect of strain, it is possible that vortices and ferroelastic domains (a/c , a_1/a_2) coexist in the PTO/STO superlattices grown on the scandate substrates such as DyScO_3 , which gives large tensile strains to the PTO/STO superlattices²⁹. Moreover, flux-closures and vortices may be essentially the same topological structure (on GdScO_3 substrate), and reducing the thickness of the superlattices can result in the transition from flux-closures to vortices^{15,18,22}. Thus in summary here, the 180° domains are possibly formed in the epitaxial single PTO films grown on the STO substrates, but the a/c and a_1/a_2 domains, and vortex topology are

unlikely from the strain aspect.

Figure R4.3 RSM analysis. **a-h** A series of RSMs in the q_x - q_z scattering plane around the (103) substrates reflection for $(\text{PTO}_n/\text{STO}_n)_{10}$ ($n = 37, 28, 23, 19, 12, 9, 4, 2$ u.c.) superlattices. Figure R4.3 is from manuscript Fig. 1.

iii: Most importantly, we have further used TEM and STEM based methods, up to the atomic scale level, to reveal the structures of the skyrmions here (Figs. 1,3 in the manuscript and supplementary Figs. 5,6,9-11,15,16 in the supplementary materials). The c/c 180° domain walls are sharp, and the c/c 180° domains are regular periodic arrangement in Figure R4.2b. The diffraction contrast images showed that the domain walls are fuzzy and do not show a regular periodic arrangement (Figure R4.4). The diffraction contrast image shows that the domains are short-range order, as previously reported. We observed the 180° domain along the [001] direction, which is distributed in periodic strips (Figure R4.2b). But the under-focus images of the planar-view shows that it has a circular character (Fig. 3, Supplementary Figs. 9-11,15,16). Polarization mapping also accords with the skyrmions polarization characteristics (Fig. 3, Supplementary Figs. 9,10,15,16). The diffraction contrast images and polarization images are consistent with the skyrmions characteristics, and it is confirmed that the domain is not $c/c180^\circ$. Moreover, in the planar-view, the a_1/a_2 domain walls are sharp along the [110] or $[1\bar{1}0]$ (Figure R4.2b), but we observed the circular domains in our work (Fig. 3, Supplementary Figs. 9-11,15,16). Therefore, what we observed in the PTO/STO superlattices grown on the STO substrates is not consistent with the a/c and a_1/a_2 domain signature. In addition, in the planar-view, vortex and flux-closure are also tubular, periodic distributions¹⁸. But we did not observe vortex with periodic tubular. Polarization analysis shows that they do not conform to the vortex characters. By the way, why the signal to noise is very poor in the RSM here in our case, is because that the periods of these skyrmions are not so perfect as those from either the vortex or a_1/a_2 domains, as noticed by the reviewer.

In summary, it is unlikely that there are c/c , a/c , a_1/a_2 and vortex in PTO layers based on RSM results. Strain analysis showed that c/c , a/c , a_1/a_2 and vortex were unlikely to form in PTO/STO superlattices grown on the STO substrates. In addition, TEM images do not reveal the characteristics of c/c , a/c , a_1/a_2 and vortex. As a result,

no complex mixed phases exist in PTO/STO superlattices grown on the STO substrates. The skyrmions and their evolutions were thus collaborated and convinced by our TEM/STEM and other characterizations. We agree the reviewer that much details may be revealed in the diffraction based technologies, and we are also trying to design related experiments in the future.

Figure R4.4 Diffraction contrast analysis and SAED. **a-h** A range of cross-sectional DF images shows thickness-dependent skyrmions evolution in the $(\text{PTO}_n/\text{STO}_n)_{10}$ ($n = 37, 28, 23, 19, 12, 9, 4, 2$ u.c.) superlattices. The illustrations in **f-h** are the partial enlarged view of the skyrmion diffraction contrast images. A set of electron diffraction spots from selected area electron diffraction (SAED) patterns are enlarged. Figure R4.4 is from manuscript Fig. 1.

Discussion: Strain and thickness

As shown in previous work and the phase-field work (e.g. Small Methods 6, 2200486 (2022)), the phases are strongly dependent on the superlattice period and the strain state.

Based on the above strain analysis ii and Supplementary Table 2, the a/c , a_1/a_2 are ferroelastic domain, the vortex and flux-closure suffer from large strain modulation. The formation and transformation of vortex and flux-closure to a/c and a_1/a_2 domains are affected by strain. a/c , a_1/a_2 are correlated with vortex, flux-closure. However, c/c is ferroelectric domain, and the ferroelectric domain is significantly affected by the electric boundary conditions. The skyrmions are also significantly affected by short-

circuit conditions (Fig. 4d, Supplementary Figs. 20,21). So the formation of the skyrmion is correlated with the c/c domain. The effect of strain on skyrmion is relatively weak, which has been confirmed by the phase-field simulation (Fig. 2, Supplementary Figs. 7,8). It is very important for the PTO/STO superlattices to grow on which substrate^{18,37,62}. This determines which strain (tensile strain, zero strain, compression strain) the PTO/STO superlattice is modulated by, thus determining the formation of ferroelastic or ferroelectric domains in PTO films.

Changes (R4.2): We added the above discussion to the supplementary note 2 and 3 on Page 4-6 in the revised supplementary materials. We added the above references omitted on Page 21 and 23 in the revised manuscript.

29. Damodaran, A. R. et al. Phase coexistence and electric-field control of toroidal order in oxide superlattices. *Nat. Mater.* **16**, 1003-1009 (2017).

62. Guo, X. W. et al. Theoretical understanding of polar topological phase transitions in functional oxide heterostructures: A review. *Small Methods* **6**, 2200486 (2022).

Comment (R4.3): A side note on TEM, it is also not clear that there is skyrmion order present everywhere. TEM only samples a tiny fraction of the sample and is known to change the strain conditions.

Reply to comment (R4.3): We thank the reviewer for this in-depth comment. The observable sample length by TEM is about 800 μm (Figure R4.5). Depending on the sample preparation, the observable length may range from millimeters to nanometers. We have prepared several TEM sample, and most importantly, we have obtained a large number of TEM images. In this work, our TEM related results are based on the large-scale image data. Finally we selected the best TEM contrast images in manuscript. We also added some additional diffraction contrast images, as shown in Figure R4.6. In this work, we never found any c/c , a/c , a_1/a_2 , or vortex signals. All the results support that what we have observed are skyrmions.

Indeed, the TEM sample preparation may change the strain conditions. Based on the above strain analysis R4.2, as well as the study of skyrmion and vortex, we believe that skyrmion is not significantly affected by the strain. Because skyrmions have the ferroelectric domain properties rather than ferroelastic domain properties. Skyrmion is different from vortex and flux-closure. Therefore, for ferroelectric skyrmion, the changed strain by TEM sample preparation can be ignored. Most importantly, the strain state in the planar-view samples here are the same as the film (Fig. 1, supplementary Figs. 5,9-11,15,16), since the substrate was kept in the TEM sample.

In addition, the single PTO/STO thickness is about 2 ~ 15 nm, and the skyrmion period is about 5 ~ 15 nm, which is very suitable for TEM characterization. Due to the limitation of resolution, the optical microscope and scanning electron microscope (SEM) can not meet the requirements. So it is very suitable for TEM to study skyrmions. During experimental operation, the screen current is controlled in a reasonable range.

Figure R4.5 Cross-sectional sample that can be used for TEM observation.

Figure R4.6 Other areas diffraction contrast images. **a-h** A range of cross-sectional DF images shows thickness-dependent skyrmions evolution in the $(\text{PTO}_n/\text{STO}_n)_{10}$ ($n = 37, 28, 23, 19, 12, 9, 4, 2$ u.c.) superlattices.

Changes (R4.3): We added three sentences on Page 17 in the Methods (TEM and STEM) in the revised manuscript. We added one supplementary Fig. 5 (Figure R4.6) on Page 10 in the revised supplementary materials.

(on Page 17 in revised manuscript).....Here, as also indicated by our phase-field calculations, the effect of electric boundary conditions on the formation of skyrmions

is much greater than that of strains. Thus the TEM sample preparation will not change the formation of the skyrmions in the superlattices grown on STO.

Comment (R4.4): The skyrmion order has a distinct reciprocal space pattern as shown in previous work. While this is a (103) peak and not (003) as shown in references 30, 31. The authors make the statement that the samples show skyrmion order for all thicknesses, but the data shows that is not true.

Reply to comment (R4.4): We thank the reviewer for this question to stimulate our deeper thinking. This is important for improving our work. We have reviewed previous literatures on the topological structure of ferroelectric films. Researchers often use high-resolution X-ray diffraction (RSM) to characterize the sample to obtain overall information. They acquired many different diffraction peaks during the study of vortex, flux-closure, and skyrmions, for example, $\bar{1}03_{pc}^{16}$, 011_{pc}^{18} , 022_{pc}^{29} , 003^{37} , $002^{16,38}$. The period of the vortex was acquired from 011_{pc} diffraction peak¹⁸. Here in our case, as also discussed in above questions, we also used TEM to characterize the microstructure. The TEM has powerful functions, such as observing morphology by diffraction contrast images (DF, BF, under-focus STEM), determining periodic structure by selected area electron diffraction (SAED) patterns, and obtaining ions displacement based on HAADF-STEM images (polarization mapping). In addition, the phase-field simulation is also a suitable method to confirm or predict the skyrmions structure in all thickness samples. In this work, for real space information, DF images, under-focus STEM images, polarization mapping combined with phase-field simulation all confirm the existence of skyrmion in different superlattices, which is consistent with the RSM and SAED results (Figs. 1-4 and supplementary Figs. 5,6,9-11,15,16). Therefore, we believe that our results are solid.

We thank again the reviewer for the insightful questions and comments to improve our work.

REVIEWER COMMENTS

Reviewer #1 (Remarks to the Author):

The revised manuscript is improved with reflecting the reviewer comments. The phenomenological origin of the deviation from the Kittel's law is now discussed in more detail. Originally, the manuscript contains many information of microscopic details as a result of the extensive and systematic works. I would like to recommend acceptance for publication in Nature Communications.

Reviewer #2 (Remarks to the Author):

The authors have sufficiently addressed all my requests for additional experimental verification to support their demonstration that the skyrmion periods and superlattice thicknesses disobey Kittel's law in PTO/STO superlattices with ultrathin PTO layers. I would like to thank the Authors for their explanations. I believe this discovery in ferroelectric topological domains is essential to research aiming at ferroelectric ultra-high density storage.

Therefore, I recommend publication in Nature Communications without further review required.

Reviewer #3 (Remarks to the Author):

NCOMMS-22-42842A: "Absence of critical thickness for polar skyrmions with breaking the Kittel's law" by F. Gong et. al

The authors have made significant efforts to improve the quality of data and discussion in the manuscript. I recommend the publication of the manuscript if the authors could make further minor revisions based on the following concerns:

1. In response to comment 1, the authors have detailed the novelty of the work. However, this is still not reflected in the manuscript. Authors are urged to improve the introduction and discussion/conclusion by contrasting their current work with the previous works to further clarify the novelty for the readers.
2. Scale bars should be added to micrographs in Figure 3a, insets of Supplementary Fig 2, Supplementary Fig 12c, and Supplementary Figure 16c.
3. The manuscript has some grammatical mistakes.

Reviewer #4 (Remarks to the Author):

This work examines the question of if Kittel's law for the scaling of FE domains is valid to be applied to the case of PTO/STO superlattices. With a combination of X-ray diffraction, TEM and phase field, the authors propose a distinct scaling of skyrmions as a function of layer thickness.

The authors have addressed some of the questions raised except for one important one, that is the understanding of the XRD patterns. I am still confused why the arrows are marking the positions of the small satellite rather than the most intense ones at $qz \sim 7.35$ in the $n=37$ to $n=19$ samples. These are the well-known skyrmion satellites that are broad along the qz direction since they are not strongly correlated between adjacent layers. I think a discussion

of scaling in this thickness range makes sense due to the similar domain structure shown by the satellites. However, $n=12$ is quite different. The sharp peaks in both q_x and q_z suggest a large coherence length uncommon to the skyrmion. As well, the large coherence along c would suggest 3D couple between the layers. For this how does one count the thickness since the domains are coherent along the full thickness of the superlattice as seen because the q_z width is the same as the specular superlattice peak? For $n=4$ and 2 , the XRD patterns do not clearly show a peak in the diffraction maps so it seems hard to add them to the scaling law. Perhaps the authors should use the scaling in the region that supports similar domain structure from $n=37$ to 19 ?

Reply to reviewers' questions and comments:

Ref number: NCOMMS-22-42842A

Title: Absence of critical thickness for polar skyrmions with breaking the Kittel's law

Authors: Feng-Hui Gong, Yun-Long Tang, Yu-Jia Wang, Yu-Ting Chen, Bo Wu, Li-Xin Yang, Yin-Lian Zhu & Xiu-Liang Ma

19 April, 2023

Reply to reviewer 1 (R1):

We appreciate the positive comments by the reviewer that “The revised manuscript is improved with reflecting the reviewer comments.”, “The phenomenological origin of the deviation from the Kittel's law is now discussed in more detail.”, and that “Originally, the manuscript contains many information of microscopic details as a result of the extensive and systematic works.”. We also thank the reviewer for the positive recommendation that “I would like to recommend acceptance for publication in Nature Communications.”.

Reply to reviewer 2 (R2):

We appreciate the positive opinions by the reviewer that “The authors have sufficiently addressed all my requests for additional experimental verification to support their demonstration that the skyrmion periods and superlattice thicknesses disobey Kittel's law in PTO/STO superlattices with ultrathin PTO layers.”, “I would like to thank the authors for their explanations.”, and that “I believe this discovery in ferroelectric topological domains is essential to research aiming at ferroelectric ultra-high density storage.”. We also appreciate the positive recommendation by the reviewer that “I recommend publication in Nature Communications without further review required.”.

Reply to reviewer 3 (R3):

We appreciate the positive comments by the reviewer that “The authors have made significant efforts to improve the quality of data and discussion in the manuscript.”. We also appreciate the positive recommendation by the reviewer that “I recommend the publication of the manuscript if the authors could make further minor revisions.”. In the meantime, the reviewer raises three questions. We fully understand the reviewer's concerns, and here we will address all the questions and discuss all the comments one-by-one in the following. The revisions were indicated in **RED** characters in the revised manuscript.

Comment (R3.1): In response to comment 1, the authors have detailed the novelty of the work. However, this is still not reflected in the manuscript. Authors are urged to improve the introduction and discussion/conclusion by contrasting their current work with the previous works to further clarify the novelty for the readers.

Reply to comment and changed (R3.1): We thank the reviewer for this meaningful suggestion. We have added related discussions in the introduction and conclusion sections to further clarify the novelty for the readers.

(on Page 2 in the revised manuscript)

...Moreover, this breakdown was theoretically believed to bring the ultrathin electronic devices with ultrahigh density storages possible⁵⁻¹³, but never confirmed in experiments.

While these works are illuminative, polar topologies, such as the skyrmions and their distributions in ultrathin PbTiO₃ (PTO) layers, were not revealed. Particularly, powered by aberration corrected (scanning) transmission electron microscopy¹⁴⁻²⁵ ((S)TEM), the discovery and study of topological polar structures make them well suitable for further exploring the Kittel's law in the ultrathin limit²⁶⁻⁴⁵...

(on Page 11 in the revised manuscript)

...Polar skyrmions can be maintained in ultrathin superlattices and bilayers with PTO layer as thin as two unit cells, which will be important for the future study of skyrmion-based ferroelectric physics and related electronic elements. The skyrmion periods (d) and superlattice thicknesses (h) disobeys the Kittel's law, especially for the samples with ultrathin PTO layers...

Comment (R3.2): Scale bars should be added to micrographs in Figure 3a, insets of Supplementary Fig 2, Supplementary Fig 12c, and Supplementary Figure 16c.

Reply to comment and changed (R3.2): We appreciate the meaningful suggestions by the reviewer. We have added the scale bars in Figure 3a, insets of supplementary Fig. 2, supplementary Fig. 12c, and supplementary Fig. 16c. We checked all the scale bars in the revised manuscript and supplementary materials.

Comment (R3.3): The manuscript has some grammatical mistakes.

Reply to comment and changed (R3.3): Thank you for your valuable comments. We have carefully checked and corrected the grammatical mistakes in the revised manuscript and supplementary materials.

Reply to reviewer 4 (R4):

We appreciate the meaningful comments by the reviewer that “This work examines the question of if Kittel's law for the scaling of FE domains is valid to be applied to the case of PTO/STO superlattices.”, and that “With a combination of X-ray diffraction, TEM and phase field, the authors propose a distinct scaling of skyrmions as a function of layer thickness.”. In the meantime, the reviewer raises specific question. We fully understand the reviewer's concerns, and here we will address this question and discuss this comment in the following. Corresponding revisions are highlighted in **RED** in the marked, revised manuscript.

Comment (R4.1): The authors have addressed some of the questions raised except for one important one, that is the understanding of the XRD patterns. I am still confused why the arrows are marking the positions of the small satellite rather than the most intense ones at $q_z \sim 7.35$ in the $n = 37$ to $n = 19$ samples. These are the well-known

skyrmion satellites that are broad along the q_z direction since they are not strongly correlated between adjacent layers. I think a discussion of scaling in this thickness range makes sense due to the similar domain structure shown by the satellites. However, $n = 12$ is quite different. The sharp peaks in both q_x and q_z suggest a large coherence length uncommon to the skyrmion. As well, the large coherence along c would suggest 3D couple between the layers. For this how does one count the thickness since the domains are coherent along the full thickness of the superlattice as seen because the q_z width is the same as the specular superlattice peak? For $n = 4$ and 2 , the XRD patterns do not clearly show a peak in the diffraction maps so it seems hard to add them to the scaling law. Perhaps the authors should use the scaling in the region that supports similar domain structure from $n = 37$ to 19 ?

Reply to comment (R4.1): We thank the reviewer again for this valuable comment and we will further discuss and revise. First, we fully agree that different PTO layer thicknesses may result in different diffraction signal details, which may reflect the further coupling of polar topologies along the out of plane direction, as suggested and noticed by the reviewer. These details probably exist in PTO/STO superlattices with different layer thicknesses and under different strains, and thus the coupling effects for polar typologies as polar vortex, polar skyrmions, etc, we believe, still need to be studied in the future (as will be mentioned in Figure R4.1 and R4.2). We have adjusted the arrows to the position of the most intense satellite peaks in the $n = 17 \sim 39$ u.c. samples. Next, we will further, systematically discuss changes in the shape of satellite peaks in X-ray diffraction (XRD) patterns. We have also carried out detailed analysis and comprehensive understanding to confirm the credibility. Here, we have the following considerations:

i: Previous results also show that the shape of the vortex satellite peaks may be slight differences. Figure R4.1a indicates the extended satellite peaks for the 16/16 superlattices²⁹. Figure R4.1b shows the circular satellite peaks for the 20/20 superlattices²⁰. However, these satellite peaks all represent vortex structures. In addition, the selected area electron diffraction (SAED) patterns also show different satellite diffraction spots^{18,20} (Figure R4.2). In fact, some experimental factors may also affect the shape of satellite peaks, such as diffraction geometry, specific X-ray optical components, the setup of each measurement, and the determination of the detection system. In our work, in addition to the information of reciprocal space, a large number of transmission electron microscopy (TEM) images were analyzed in detail to study the microstructures of the superlattices, therefore, the scaling law here can be supported by the combination of these two complementary methods.

Figure R4.1 Synchrotron-based RSMs for superlattices containing vortices. The data pictures are from reference^{20,29}.

Figure R4.2 The SAED patterns from the areas including the vortices and substrates. The data pictures are from reference^{18,20}.

ii: We also make some other considerations about the possible reasons for the different shapes of skyrmions satellite peaks. The elongated satellite peaks of the thicker superlattice ($n = 17 \sim 39$ u.c.) may be caused by the inhomogeneous distribution and morphology of skyrmions, which was further confirmed by under-focus high-angle annular dark-field (HAADF) STEM images (supplementary Figs. 9, 11). In thicker superlattice, skyrmions are closer to the cylinders or elongated cylinders (Fig. 1, supplementary Figs. 9, 11, 14). And the circular satellite peaks imply that more ordered and regular skyrmions exist in the $\text{PTO}_{12}/\text{STO}_{12}$ superlattices, which was also further confirmed by under-focus HAADF-STEM images (supplementary Figs. 10, 11). As the single PTO/STO thickness decreases, the skyrmions shrink, skyrmions are closer to the spheres (Fig. 1, supplementary Figs. 10, 11, 14).

iii: There may be another factor for the shape of the satellite peaks. In real space, the thicker the single PTO/STO layer for superlattice, in reciprocal space, the closer the distance between Bragg diffraction spots (0, 1, 2 order). This also applies to satellite peaks. The result is that some Bragg (or satellite) peaks of thicker superlattice display some overlap-effects in the RSM.

Indeed, we think the reviewer's discussion is very meaningful and helpful. The sharp peaks in both q_x and q_z might suggest a large coherence length. As well, the large coherence along c may suggest possible 3D couple between the layers. However, in the dark-field (DF) images (Fig. 1, supplementary Fig. 5), intensity modulation exists in the PTO layer, the contrast is almost uniform in the STO layer. We have also grown $(\text{PTO}_n/\text{STO}_n)_1$ ($n = 50, 23, 12, 2$ u.c.) bilayers on the STO substrates (supplementary

Figs. 15, 16). The results show that they all conform to the skyrmions characteristics.

As noticed, the diffraction maps ($n = 4, 2$ u.c.) show a weak signal. Nevertheless, we have added a large amount of TEM data to support our results (Figs. 1, 3, supplementary Figs. 5, 6, 9-11, 14-16). Based on a series of TEM results for $(\text{PTO}_n/\text{STO}_n)_{10}$ superlattices and $(\text{PTO}_n/\text{STO}_n)_1$ bilayers, skyrmions exist not only in ultrathin superlattices, but also in ultrathin bilayers. In our works, by real-space diffraction contrast images, SAED patterns, atomic scale polarization maps, and so on, we also obtained the period of polar skyrmions, which is consistent with the period of the RSM.

Finally, as suggested by the reviewer, we believe that more works still need to be done in the future to reveal the out of plane coupling effects of polar topologies and their possible evolutions.

Changes (R4.1): We added the above discussion to the supplementary note 1 on Page 3 in the revised supplementary materials.

(on Page 3 in the revised supplementary materials)

Here, we will systematically discuss changes in the shape of satellite peaks in XRD patterns. Previous results also show that the shape of the vortex satellite peaks may be slight differences (the extended satellite peaks for the 16/16 superlattices²⁹ and the circular satellite peaks for the 20/20 superlattices²⁰). However, these satellite peaks all represent vortex structures. In addition, the SAED patterns also show different satellite diffraction spots^{18,20}. In fact, some experimental factors may also affect the shape of satellite peaks, such as diffraction geometry, specific X-ray optical components, the setup of each measurement, and the determination of the detection system. In our work, in addition to the information of reciprocal space, a large number of TEM images were analyzed in detail to study the microstructures of the superlattices, therefore, the shape of the satellite peaks may not directly affect our discussion of the scaling law.

We also make some other considerations about the possible reasons for the different shapes of skyrmions satellite peaks. The elongated satellite peaks of the thicker superlattice ($n = 17 \sim 39$ u.c.) may be caused by the inhomogeneous distribution and morphology of skyrmions, which was further confirmed by under-focus HAADF-STEM images (supplementary Figs. 9, 11). And the circular satellite peaks imply that more ordered and regular skyrmions exist in the $\text{PTO}_{12}/\text{STO}_{12}$ superlattices, which was also further confirmed by under-focus HAADF-STEM images (supplementary Figs. 10, 11).

REVIEWERS' COMMENTS

Reviewer #4 (Remarks to the Author):

This work examines the question of if Kittel's law for the scaling of FE domains is valid to be applied to the case of PTO/STO superlattices. With a combination of X-ray diffraction, TEM and phase field, the authors propose a distinct scaling of skyrmions as a function of layer thickness.

The authors have addressed my concerns to the best of their ability.

Reply to reviewers' comments:

Ref number: NCOMMS-22-42842B

Title: Absence of critical thickness for polar skyrmions with breaking the Kittel's law

Authors: Feng-Hui Gong, Yun-Long Tang, Yu-Jia Wang, Yu-Ting Chen, Bo Wu, Li-Xin Yang, Yin-Lian Zhu & Xiu-Liang Ma

22 May, 2023

Reply to reviewer 4:

We appreciate the comments by the reviewer that “This work examines the question of if Kittel’s law for the scaling of FE domains is valid to be applied to the case of PTO/STO superlattices.”, “With a combination of X-ray diffraction, TEM and phase field, the authors propose a distinct scaling of skyrmions as a function of layer thickness.”, and that “The authors have addressed my concerns to the best of their ability.”.